# TKI-mediated inhibition of NLRP1 inflammasome restores erythropoiesis in DBA syndrome

Juan M Lozano-Gil [1,2,3], Lola Rodríguez-Ruiz [1,2,3], Manuel Palacios [3,4,5], Jorge Peral[3,4,5], Susana Navarro[3,4,5], José L Fuster [2,6], Cristina Beléndez[3,7], Andrés Jérez[8], Laura Murillo-Sanjuán[3,9], Cristina Díaz-de-Heredia [3,9], Guzmán López-de-Hontanar [3,10], Josune Zubicaray[3,10], Julián Sevilla[3,10], Francisca Ferrer-Marín[3,11], María P Sepulcre[1,2,3], María L Cayuela [2,3], Diana García-Moreno [2,3], Alicia Martínez-López [2,3✉], Sylwia D Tyrkalska [2,3✉] & Victoriano Mulero [1,2,3✉]

## Abstract

**Diamond-Blackfan anemia syndrome (DBAS) is marked by defective erythropoiesis caused by impaired ribosome biogenesis and aberrant signaling. Here, we investigate how ribosomal stress-induced activation of the NLRP1 inflammasome affects erythroid differentiation in DBAS. We demonstrate that FDA/EMA-approved tyrosine kinase inhibitors (TKIs) effectively mitigate defective erythropoiesis by inhibiting NLRP1 inflammasome activation. In K562 cells, nilotinib suppresses the ZAKα/P38/NLRP1/CASP1 axis, leading to increased GATA1 levels and upregulation of key erythroid genes. These effects were validated in human CD34+ hematopoietic stem and progenitor cells (HSPCs) and zebrafish models, where nilotinib, imatinib, and dasatinib promoted erythropoiesis while reducing caspase-1 activity. In Rps19-deficient zebrafish, RPS19-deficient human HSPCs, and HSPCs from DBAS patients, TKIs rescued erythroid differentiation and restored hemoglobin levels. Our findings highlight that targeting the NLRP1 inflammasome with TKIs may provide a novel therapeutic strategy for DBAS and other ribosomopathies.**

**Keywords** NLRP1; Tyrosine Kinase Inhibitors; Zebrafish; Drug Repurposing; Diamond-Blackfan Anemia Syndrome
**Subject Categories** Haematology; Immunology

## Introduction

Initially identified as key regulators of innate immunity and inflammation, inflammasomes have recently been implicated in a broader range of biological processes, including hematopoiesis (Rodriguez-Ruiz et al, 2020). Furthermore, while the expression of inflammasome components was first reported in macrophages, granulocytes, and dendritic cells (Evavold and Kagan, 2019; Martinon et al, 2002), and subsequently in lymphocytes (Martin et al, 2016; Phan et al, 2007), emerging evidence demonstrates that hematopoietic stem and progenitor cells (HSPCs) also express inflammasome components (Lenkiewicz et al, 2019; Ratajczak et al, 2020; Rodriguez-Ruiz et al, 2023; Tyrkalska et al, 2019). HSPCs not only express inflammasome components but also rely on these components, including the well-characterized NLR family pyrin domain containing 3 (NLRP3) inflammasome, for the regulation and modulation of hematopoiesis. The effects of NLRP3 on HSPCs are mediated through both direct and indirect pathways. On the one hand, NLRP3 activation influences the development and proliferation of HSPCs within the hematopoietic tissue (Frame et al, 2018; Ratajczak et al, 2020). On the other hand, it regulates their migration from the bone marrow into the peripheral blood in response to pharmacological mobilization or stress (Lenkiewicz et al, 2019; Ratajczak et al, 2019). Additionally, NLRP3 plays a role in guiding the reintegration of HSPCs into the bone marrow following transplantation and contributes to the aging process of these cells (Adamiak et al, 2020). Moreover, inflammasome activation within the bone marrow microenvironment is thought to modulate the balance of distinct hematopoietic cell populations. Although there is no direct evidence that NLRP3 in HSPCs regulates their differentiation, the involvement of downstream inflammasome components, such as apoptosis-associated speck-like protein containing a caspase recruitment domain (ASC) and caspase-1 (CASP1), in this process suggests that other NLRs may be implicated instead (Tyrkalska et al, 2019). Regardless, dysregulation of the NLRP3 inflammasome is closely associated with impaired myelopoiesis and lymphopoiesis, contributing to the development of hematologic disorders, including myelodysplastic syndromes,

[1]Departamento de Biología Celular e Histología, Facultad de Biología, Universidad de Murcia, 30100 Murcia, Spain. [2]Instituto Murciano de Investigación Biosanitaria (IMIB-Pascual Parrilla), 30120 Murcia, Spain. [3]Centro de Investigación Biomédica en Red (CIBERER), ISCIII, 28029 Madrid, Spain. [4]Centro de Investigaciones Energéticas, Medioambientales y Tecnológicas (CIEMAT), 28040 Madrid, Spain. [5]Unidad de Terapias Avanzadas, IIS-Fundación Jiménez Díaz (IIS-FJD, UAM), Madrid, Spain. [6]Sección de Oncohematología Pediátrica, Servicio de Hematología, Hospital Clínico Universitario Virgen de la Arrixaca, 30120 Murcia, Spain. [7]Hospital General Universitario Gregorio Marañón, Instituto Investigación Sanitaria Gregorio Marañón, Facultad de Medicina Universidad Complutense, ERN-EuroBloodNet, 28007 Madrid, Spain. [8]Departamento de Hematología y Oncología Clínica, Hospital General Universitario Morales Meseguer, 30008 Murcia, Spain. [9]División de Hematología y Oncología Pediátrica, Hospital Universitario Vall d´Hebron, Barcelona 08035, Spain. [10]Pediatric Hematology and Oncology Department, Hospital Infantil Universitario Niño Jesús, 28009 Madrid, Spain. [11]Fundación Jiménez-Díaz, IIS-FJD, Madrid 28015, Spain. ✉E-mail: alicia.martinez34@um.es; tyrkalska.sylwia@gmail.com; vmulero@um.es

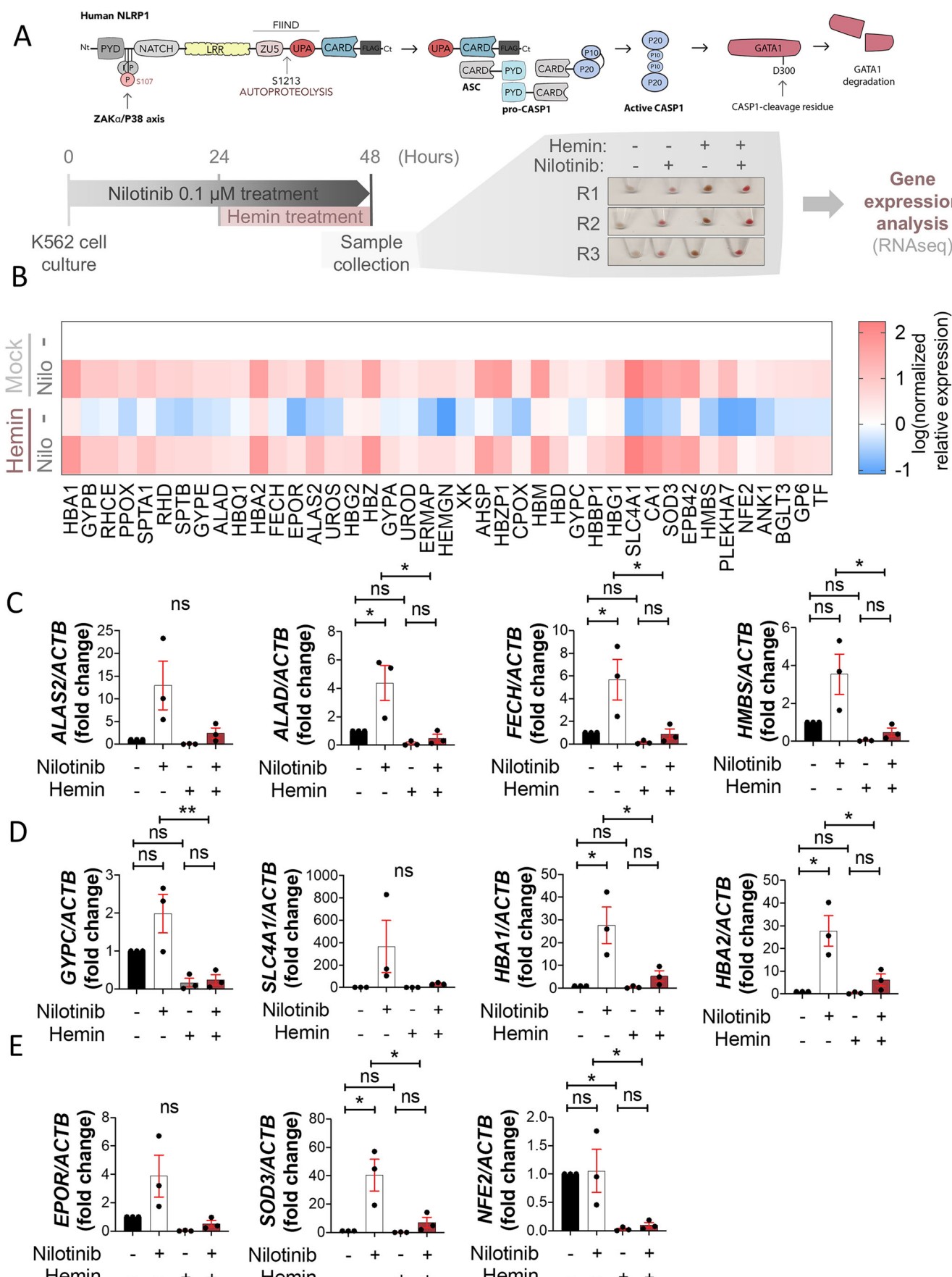

Figure 1. Nilotinib induces the expression of key erythroid genes in K562 cells.

(A) K562 cells were pretreated with 0.1 μM nilotinib, followed by differentiation with 50 μM hemin for 24 h before transcriptomic analysis. (B) Heatmap showing the top upregulated erythroid genes upon nilotinib treatment. (C–E) RT–qPCR analysis of select upregulated erythroid genes involved in structural components of erythrocytes (C), heme biosynthesis (D), and other erythropoietic functions (E) following nilotinib treatment. Data are shown as the mean ± SEM ($N = 3$). $P$ values were calculated using one-way ANOVA and Tukey's multiple range test, ns, non-significant; **$p < 0.01$, ***$P < 0.001$ and ****$P < 0.0001$. (C) ALAD: 0_DMSO wrt 0_NILO $p = 0.0296$*, 0_NILO wrt 24_NILO $p = 0.0158$*, FECH: 0_DMSO wrt 0_NILO $p = 0.0451$*, 0_NILO wrt 24_NILO $p = 0.0405$*, HMBS: 0_NILO wrt 24_NILO $p = 0.0294$*; (D) GYPC 0_NILO wrt 24_NILO $p = 0.0096$**, HBA1: 0_DMSO wrt 0_NILO $p = 0.0193$*, 0_NILO wrt 24_NILO $p = 0.0418$*, HBA2: 0_DMSO wrt 0_NILO $p = 0.0107$*, 0_NILO wrt 24_NILO $p = 0.0287$*; (E) SOD3: 0_DMSO wrt 0_NILO $p = 0.0128$*, 0_NILO wrt 24_NILO $p = 0.0273$*, NFE2: 0_DMSO wrt 24_DMSO $p = 0.0465$*, 0_NILO wrt 24_NILO $p = 0.0497$*. Source data are available online for this figure.

myeloproliferative neoplasms, leukemia, and graft-versus-host disease after transplantation (He et al, 2016).

Our group was the first to report that the canonical inflammasome of HSPCs orchestrates the erythroid vs. myeloid lineage decision by cleaving and inactivating the master erythroid transcription factor GATA1 (Tyrkalska et al, 2019). Although this mechanism is conserved in zebrafish and humans, the specific NLR sensor involved and its activation mechanism remained unknown until recently, when we demonstrated that NLRP1 orchestrates this regulation and is activated in HSPCs via phosphorylation of S107 in its linker domain by the ZAKα/P38 axis following erythroid differentiation-induced ribotoxic stress (Rodríguez-Ruiz et al, 2023). This mechanism of activation highlighted the potential of inhibiting ZAKα using FDA/EMA-approved tyrosine kinase inhibitors (TKIs), such as nilotinib, as a promising therapeutic strategy to prevent excessive or pathological NLRP1 activation in ribosomopathies, such as Diamond-Blackfan anemia syndrome (DBAS). The primary cause of DBAS are germline heterozygous loss-of-function mutations in small- or large-subunit ribosomal protein genes, resulting in defective ribosome biogenesis and/or function. Among these, mutations in RPS19 are the most prevalent, accounting for approximately 25% of cases (Da Costa et al, 2020). In this study, we tested several TKIs for their ability to modulate NLRP1 activation and demonstrated their therapeutic efficacy in restoring erythropoiesis in models of DBAS, paving the way for their repurposing in the treatment of ribosomopathies.

## Results

### Nilotinib promotes erythroid differentiation of K562 cells and CD34+ HSPCs

We first aimed to further investigate the effect of nilotinib, which we previously showed to reduce ZAKα and P38 activation (Rodriguez-Ruiz et al, 2023), on the progression of hemin-induced erythroid differentiation and GATA1 accumulation in K562 cells (Fig. 1A). We performed a RNA-seq analysis of K562 cells upon their differentiation with hemin in the presence of nilotinib and found that it was able to induce the transcript levels of several genes encoding major factors involved in erythroid maturation and function, such as heme biosynthesis (ALAS2, ALAD, FECH and HMBS), the receptor of erythropoietin (EPOR), structural molecules located in the plasma membrane of erythrocytes (GYPC and SLC4A1), globin subunits (HBA1 and HBA2), and antioxidant system (SOD3 and NFE2) (Fig. 1A,B, Datasets EV1 and EV2). These results were confirmed by RT–qPCR

(Fig. 1C–E). Curiously, nilotinib acted as a better inducer of erythroid differentiation in K562 cells than hemin, promoting a strong expression of the above-mentioned genes without hemin treatment (Fig. 1C–E).

The results obtained with nilotinib in K562 cells prompted us to investigate if nilotinib was also able to promote erythroid differentiation of HSPCs. With this aim, human cord blood CD34+ cells were cultured in differentiation media containing EPO in the presence of either DMSO or nilotinib. To dissect whether nilotinib acts primarily at the onset of erythroid commitment or can also reinforce differentiation at later stages, we added the drug either at 3–7 days post-differentiation (dpd) or at a later time point (7–10 dpd). During this process, CD34+ cells are expected to abandon their multipotency and start expressing the receptor of transferrin (CD71) at the CFU-E stage. Then, a co-expression of glycophorin A (CD235A) together with CD71 marker was expected in erythroblasts until the CD71 expression was downregulated in reticulocytes/erythrocytes. Nilotinib was able to inhibit CASP1 activity and robustly increase the transcript levels of GATA-1-dependent genes, the percentage of GATA1high cells and the percentage of erythroblasts (CD71+/CD235A+ cells) (Figs. 2A–D and EV1A–C), suggesting that nilotinib promotes erythropoiesis.

We repeated the experiments adding nilotinib at 7 dpd and, in this case, nilotinib also inhibited CASP1 activity but the effect in promoting differentiation was weaker than in the previous experiment (Fig. 2E–G). Importantly, the inhibition of CASP1 activity by nilotinib in both experiments confirmed that it was inhibiting the ZAKα/P38/NLRP1 inflammasome. All these results suggest that nilotinib induces the progression of erythroid differentiation, which may be of clinical interest since some types of congenital anemia do not only show decreased number of HSPCs but also defective erythropoiesis, for example DBAS (Da Costa et al, 2020).

### Several FDA/EMA-approved TKIs alleviate defective erythropoiesis in a zebrafish model of DBAS

Considering our previous results showing the impact of nilotinib over erythroid differentiation in human cells, we next tested whether other TKIs were able to show the same effects in zebrafish hematopoiesis. TKIs were developed for the treatment of Philadelphia chromosome (Ph)-positive chronic myeloid leukemia (CML), which is driven by the chimeric BCR-ABL gene. The first-generation drug imatinib showed an incredibly strong response in Ph-positive CML patients increasing their survival rate and life expectancy (Cortes et al, 2021; Kronick et al, 2023). Due to the

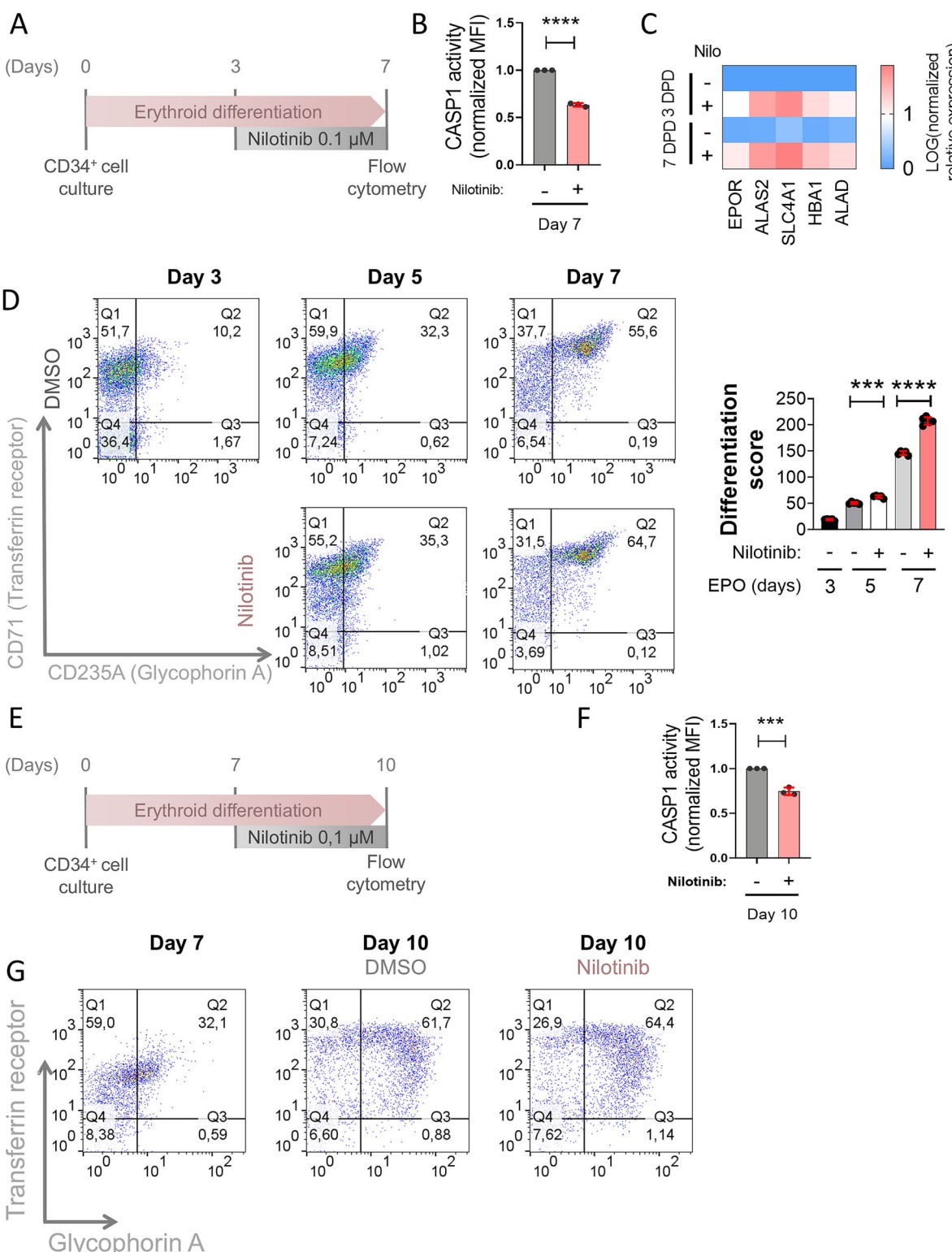

**Figure 2. Nilotinib promotes erythropoiesis of human HSPCs from healthy donors.**

(A, E) Primary human CD34+/CD133+ HSPCs from healthy donors were cultured with EPO, and 0.1 μM nilotinib was added from days 3 to 7 (A) or days 7 to 11 (E). (B, F) CASP1 activity was measured by flow cytometry with FAM FLICA and normalized to untreated control cells. (D, G) Erythroid differentiation was assessed by flow cytometry after staining with anti-CD235A-APC (Glycophorin A) and anti-CD71-FITC (Transferrin Receptor). Representative dot plots of differentiation stages are shown. The differentiation score was calculated as the ratio between CD235A+/CD71+ (intermediate erythroid progenitors) and CD235A-/CD71+ (early erythroid progenitors). MFI, mean fluorescence intensity. Data are shown as the mean ± SEM (B, F: $N = 3$; D: $N = 5$; G: $N = 2$). P values were calculated using one-way ANOVA and Tukey's multiple range test. ns, non-significant; **$p < 0.01$, ***$P < 0.001$ and ****$P < 0.0001$. (B) DMSO_D7 wrt NILO_D7 $p < 0.0001$****, (D) DMSO_D5 wrt NILO_D5 $p = 0.002$***, DMSO_D7 wrt NILO_D7 $p < 0.0001$****, (F) DMSO_D10 wrt NILO_D10 $p = 0.0005$***. Source data are available online for this figure.

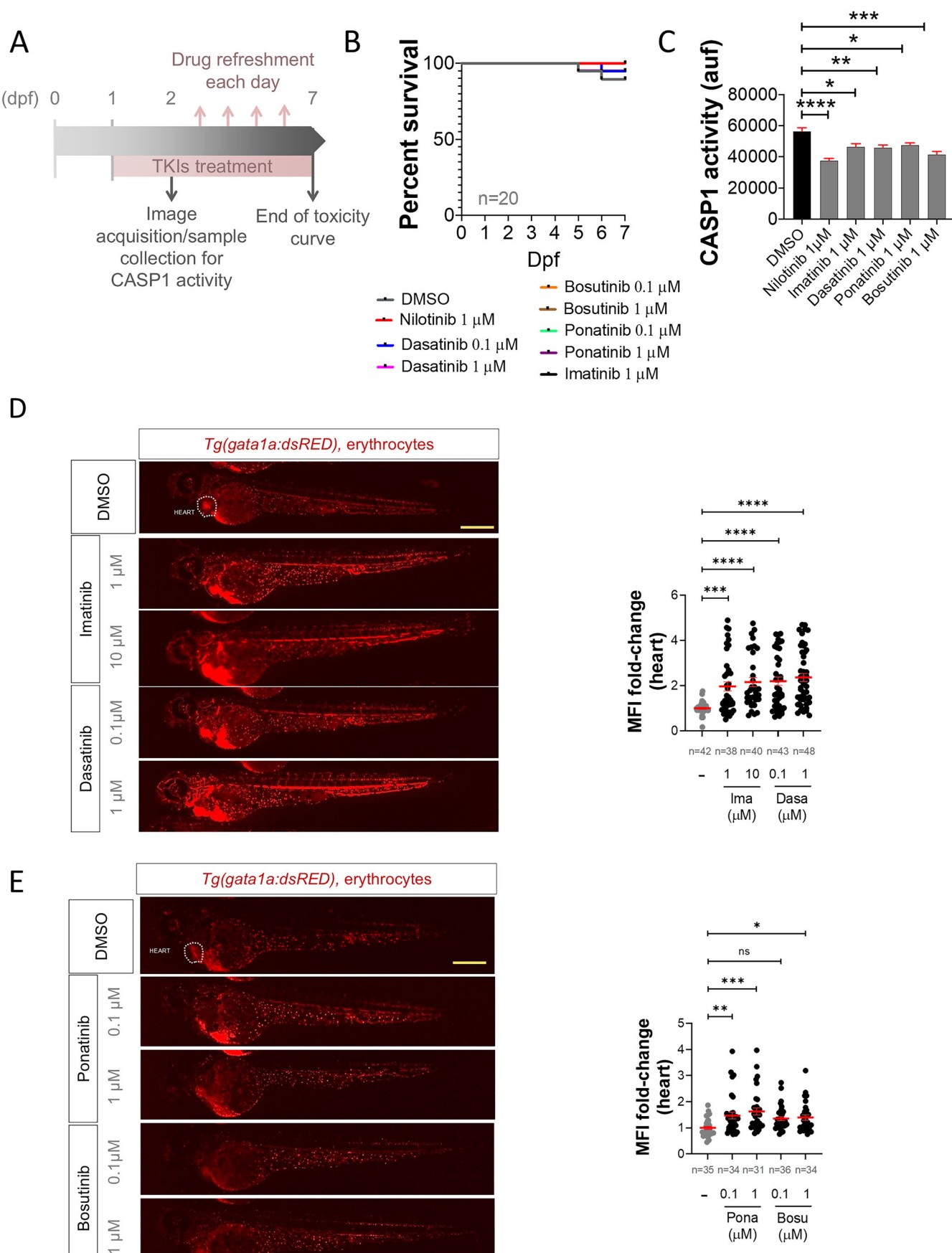

◄ **Figure 3. ZAKα inhibitors mimic the effects of nilotinib on hematopoiesis in zebrafish.**

(A) Schematic representation of experimental design. Zebrafish larvae were treated with the indicated concentrations of different TKIs from 1 to 7 days post-fertilization (dpf). (B) Survival rates of zebrafish larvae after TKI treatment at the specified doses, monitored from 1 to 7 dpf. (C–E) Effects of TKIs on hematopoiesis in zebrafish larvae. Caspase-1 activity (C) and the number of erythrocytes at the heart (D, E) were evaluated in 2 dpf larvae treated via bath immersion with the indicated doses of TKIs from 1 to 2 dpf. Representative images of erythrocytes are shown. The region of interest is shown. Each dot represents one individual and the mean ± SEM for each group is also shown (C: $N = 3$; D, E: $N$ is indicated in the figures). $P$ values were calculated using one-way ANOVA and Tukey's multiple range test (C–E) or log-rank test with Bonferroni correction (B). ns, non-significant; $**p < 0.01$, $***P < 0.001$ and $****P < 0.0001$. (C) DMSO wrt: NILO $p < 0.0001****$, IMA $p = 0.0124*$, DASA $p = 0.0085**$, IMA $p = 0.0240*$ or BOSU $p = 0.0005***$, (D) DMSO wrt: IMA_1 $p = 0.004***$ or rest of the groups $p < 0.0001****$, (E) DMSO wrt: PONA_0.1 $p = 0.0062**$. PONA_1 $p = 0.0002***$ or BOSU_1 $p = 0.0374*$. MFI, mean fluorescence intensity. Source data are available online for this figure.

resistance of imatinib during CML treatment, the second generation of TKIs were developed including bosutinib (Golas et al, 2003), dasatinib (Shah et al, 2004), nilotinib (Weisberg et al, 2005) and ponatinib (O'Hare et al, 2009). In our previous study, nilotinib robustly inhibited the Zaka/P38/Nlrp1 inflammasome in zebrafish larvae resulting in increased erythrocyte counts, decreased myeloid counts and reduced caspase-1 activity (Rodriguez-Ruiz et al, 2023). We tested imatinib, dasatinib, ponatinib and bosutinib and none of the drugs affected development and survival (Fig. 3A,B). Notably, all TKIs reduced caspase-1 activity and increased erythrocyte counts at the expense of neutrophils (Figs. 3C–E and 4A,B). Strikingly, TKI treatment of the zebrafish Spint1a-deficient model, which is characterized by Nlrp1-driven neutrophilic inflammation (Rodriguez-Ruiz et al, 2023; Tyrkalska et al, 2019) (Fig. 4C), alleviated neutrophilia without affecting neutrophil infiltration of inflamed skin (Fig. 4D).

These promising findings led us to explore the therapeutic potential of inhibiting the Zaka/Nlrp1 inflammasome with TKIs in the context of DBAS. To this end, we generated a zebrafish model of DBAS through genetic disruption of *rps19* using CRISPR-Cas9 technology (Rodriguez-Ruiz et al, 2025), as *RPS19* mutations are the most common genetic alterations in DBAS and are associated with defective erythropoiesis (Da Costa et al, 2020). Genetic experiments confirmed that the Zaka/Nlrp1 inflammasome drives anemia in *Rps19*-deficient larvae, as inhibition of either Nlrp1 or Zaka fully rescued the anemia in this model (Fig. 4A–F). Remarkably, treatment with nilotinib, dasatinib, and imatinib, but not ponatinib and bosutinib, partially rescued the severe anemia observed in *Rps19*-deficient larvae, as assessed by the hemoglobin content (Fig. 4G), highlighting the potential of these drugs as therapeutic options for DBAS.

## TKIs promote erythroid differentiation of K562 cells by inhibiting the ZAKα/NLRP1 inflammasome and stabilizing GATA1 protein

Given that our initial validation in zebrafish larvae demonstrated that other TKIs played a significant role in hematopoiesis, with effects like those previously observed with nilotinib, we next investigated whether these drugs regulated hematopoiesis in human cells through the inhibition of the ZAKα/NLRP1 inflammasome. Previously, we showed that erythroid differentiation of K562 cells with hemin resulted in the phosphorylation of ZAKα and P38, leading to NLRP1 inflammasome activation and GATA1 cleavage by CASP1 (Fig. 1A). Notably, nilotinib blocked this activation and promoted GATA1 accumulation (Rodríguez-Ruiz et al, 2023). Consistent with these findings, we observed that the first-generation TKI imatinib, at 1 μM but not at 0.1 μM, induced GATA1 protein accumulation in K562 cells undergoing hemin-induced erythroid differentiation, while

suppressing ZAKα/P38 phosphorylation. Similarly, both doses of dasatinib tested effectively replicated the effects of nilotinib (Fig. 5A,B). Interestingly, although ponatinib and bosutinib treatments exhibited a dose-dependent inhibitory effect on ZAKα/P38 phosphorylation, all tested concentrations led to GATA1 protein accumulation and promoted erythroid differentiation, as evidenced by hemoglobin accumulation and the intense red color of the cell pellets (Fig. 5A–C). Because ZAKα is known to activate JNK independently of NLRP1 phosphorylation and assembly, we also analyzed JNK1/2 activation during erythroid differentiation. Our results showed robust phosphorylation of both isoforms, which was fully suppressed by nilotinib, dasatinib, and imatinib (Fig. 5A,C). Consistent with all these results, inhibition of the ZAKα/NLRP1 axis by the different TKIs impaired CASP1 activation during erythroid differentiation of K562 cells (Fig. 5A,D). These findings suggest that GATA1 accumulation could be a consequence of CASP1 inhibition mediated by TKIs, since GATA1 is cleaved and inactivated by CASP1 (Tyrkalska et al, 2019).

## Nilotinib, imatinib and dasatinib alleviates defective erythroid differentiation of RPS19-deficient CD34+ HSPCs

To demonstrate the usefulness of nilotinib as a treatment for DBAS, we generated a DBAS model in human CD34+ cells by editing *RPS19* gene with CRISPR-Cas9 (Bhoopalan et al, 2023), reaching 35% of indel score after 13 days post-transfection (Fig. 6A). RPS19 deficiency led to an increased accumulation of BFU-E (CD71+/CD235A− cells) that failed to progress into erythroblasts (CD71+/CD235A+ cells), whereas nilotinib restored normal differentiation (Fig. 6B–D). Strikingly, RPS19 deficiency also led to enhanced CASP1 activity, which was restored to normal levels by nilotinib (Fig. 6E). Consistent with these findings, both imatinib (Figs. EV1, EV2A–D and EV3A–C) and dasatinib (Figs. EV1, EV2E–K and EV3E–J) replicated the effects of nilotinib on both wild-type and RPS19-deficient human CD34+ cells, when added either at early or late stages of erythroid differentiation. Collectively, these results suggest that enhanced ribotoxic stress of RPS19-deficient erythroid progenitors results in hyperactivation of the ZAKα/P38/NLRP1/CASP1 inflammasome leading to exacerbated degradation of GATA1 and impaired erythroid differentiation, which can be effectively counteracted by TKIs such as nilotinib, imatinib and dasatinib.

## TKIs alleviates defective erythroid differentiation of HSPCs from DBAS patients

We next examined whether the different TKIs tested in zebrafish, K562 cells and the *RSP19*-edited model of DBAS, were also able to

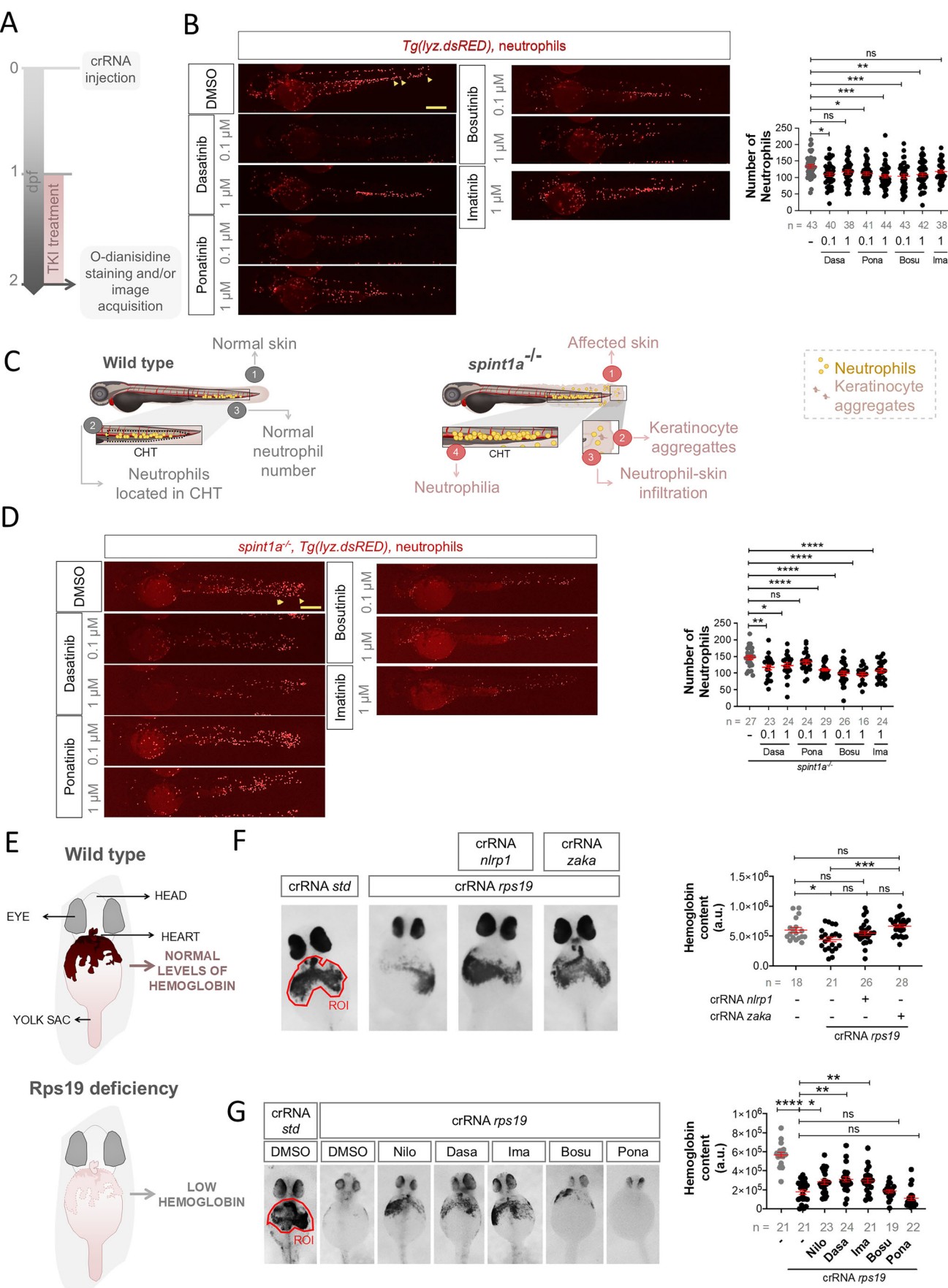

**Figure 4.   Zaka/Nlrp1 inhibition rescue anemia in zebrafish models of DBAS.**

(A, C, E) Schematic representations of experimental designs. Zebrafish larvae were treated via bath immersion with the indicated TKIs from 1 to 2 days post-fertilization (dpf). (B, D) Quantification of neutrophils and representative images of treated larvae showing neutrophils (arrowheads). (F, G) Hemoglobin staining and quantification in Rps19-deficient zebrafish larvae generated by injecting embryos with gRNA/Cas9 complexes. Representative images of hemoglobin staining in treated larvae are shown. Regions of interest (ROI) are indicated in the images. Each dot represents one individual and the mean ± SEM for each group is also shown. $P$ values were calculated using one-way ANOVA and Tukey's multiple range test (B, D, F, G). ns, non-significant; $**p < 0.01$, $***P < 0.001$ and $****P < 0.0001$. (B) DMSO wrt: DASA_01 $p = 0.0162*$, PONA_01 $p = 0.0317*$, PONA_1 $p = 0.0009***$, BOSU_01 $p = 0.009***$, BOSU_1 $p = 0.0097**$, (D) DMSO wrt: DASA_01 $p = 0.0039**$, DASA_1 $p = 0.0317*$ or rest of the groups $p < 0.0001****$, (F) CONTROL wrt crRNA_rps19 $p = 0.0286*$, crRNA_rps19 wrt crRNA_rps19-zaka $p = 0.0002***$, (G) crRNA_STD wrt crRNA_rps19 $p < 0.0001****$, crRNA_rps19 wrt: Nilo $p = 0.0148*$, Dasa $p = 0.0012**$, Ima $p = 0.0055**$. Source data are available online for this figure.

alleviate defective erythropoiesis of HSPCs from DBAS patients. To do so, mononuclear cells from bone marrow aspirates (BMMCs) or peripheral blood (PBMCs) were isolated by isopycnic centrifugation and seeded in methylcellulose medium containing EPO and colony-stimulating factors (CSFs). The results showed that all TKI tested, but ponatinib, strongly increased the number of BFU-Es derived from HSPCs of DBAS patients (Fig. 7A–G). Importantly, although the effects of TKIs on the number of CFU-GMs varied greatly among patients and were not statistically significant overall, the highest dose of nilotinib tested (0.1 μM) was also able to significantly increase the number of myeloid colonies (Fig. 7H–M). These results suggest that the inhibition of the ZAKα/P38/NLRP1/CASP1 axis with TKIs may alleviate the impaired erythropoiesis of patients suffering from DBAS and, therefore, they are attractive to be repurposed for this disease.

## Discussion

The importance of the inflammasome as therapeutic targets for the treatment of pathologies has gained interest in the clinic during the last years due to their recently reported strong implications in several diseases. Nowadays, most of FDA/EMA-approved drugs are designed to reach inflammasome-related targets, such as IL-1B and IL-18, instead of reaching directly the inflammasomes (Yao et al, 2024). Direct inhibition of specific inflammasomes components has also shown promising results in animal disease models. For instance, NLRP3 inhibitors, including MCC950, have been successfully used in Alzheimer's disease by reducing memory deficits, inflammasome activity and the levels of Aβ plaques in APP/PS1 and TgCRND8 mice, meanwhile indirect inhibitors targeting either upstream or downstream proteins, such as the caspase-1 inhibitor VX-765 or the GSDMD inhibitor disulfiram decreased IL-1B levels and improved cognitive abilities in 5×FAD mice (Ravichandran and Heneka, 2024). Tranilast, a drug that inhibits NLRP3 assembly by binding to its NACHT domain, approved to treat allergies, has shown a promising effect in vitiligo improving melanogenesis and melanosome translocation via attenuating IL-1B secretion in keratinocytes (Zhuang et al, 2020). Caffeic acid phenethyl ester, which directly binds ASC, can be used to prevent NLRP3-ASC interactions triggered by monosodium urate (MSU) crystals and, therefore, has been proposed for the treatment of acute gout (Lee et al, 2016).

Recent studies, including some from our group, have extended the relevance of the inflammasome to hematological diseases (Ratajczak et al, 2020; Rodriguez-Ruiz et al, 2020). Our group initially reported that CASP1 was able to directly cleave and

promote the degradation of the master erythroid transcription factor GATA1, fine-tuning directly GATA1 and indirectly SPI1 levels to control the myeloid-erythroid lineage decision of HSPCs (Tyrkalska et al, 2019). More recently, we identified the NLRP1 inflammasome being responsible for the activation of CASP1 in HSPCs, its strict negative regulation by LRRFIP1 and FLII in these cells, and its activation by phosphorylation of its linker domain by the ZAKα/P38 kinase axis upon ribosomal stress (Rodríguez-Ruiz et al, 2023). This discovery has revealed ZAKα as a druggable target using FDA/EMA-approved TKIs for the treatment of congenital anemias, such as DBAS, where ribosome mutations lead to both ribosomal stress and inefficient GATA1 translation (Ludwig et al, 2014). Thus, we have shown that nilotinib was able to block the activation of NLRP1 inflammasome by ZAKα/P38 kinase signaling pathway resulting in reduced CASP1 activity, and enhanced GATA1 accumulation and erythroid differentiation of K562 cells. We extend here these observations by showing that treatment of K562 cells with nilotinib robustly promotes erythroid differentiation by increasing the transcript levels of genes encoding structural and functional genes required for erythropoiesis, such as those involved in iron acquisition, hemoglobin synthesis, erythrocyte structure, and antioxidant system. Given that these genes are regulated by GATA1, and that inhibition of the ZAKα/NLRP1/CASP1 axis by nilotinib leads to increased GATA1 protein levels, it is likely that nilotinib induces their expression indirectly through GATA1 stabilization. Notably, we also observed robust activation of JNK1/2 during erythroid differentiation of K562 cells, which was fully blocked by TKIs. In keratinocytes exposed to UV light, JNK downstream of ZAKα does not activate NLRP1 but rather induces apoptosis (Robinson et al, 2022; Vind et al, 2024). Therefore, the potential contribution of JNK inhibition by TKIs to erythroid differentiation remains to be clarified.

The usefulness of nilotinib, imatinib and dasatinib to be repurposed to treat DBAS was further confirmed using a RPS19-edited human model of DBAS where they not only restored impaired erythroid differentiation of RPS19-deficient HSPCs but also exacerbated CASP1 activity, likely resulting from increased ribosomal stress (Khajuria et al, 2018). Furthermore, nilotinib, imatinib, dasatinib and bosutinib, all alleviated defective erythropoiesis of HSPCs from DBAS patients harboring different ribosome mutations, and phenocopied the effect of nilotinib in K562 cells and zebrafish; that is they (i) inhibited ZAKα/P38/NLRP1/CASP1 in K562 cells leading to increased GATA1 levels and erythroid differentiation, (ii) enhanced erythropoiesis at the expenses of granulopoiesis in zebrafish larvae, and (iii) ameliorated neutrophilia in a zebrafish model of neutrophilic inflammation. Therefore, these TKIs are promising drugs for the treatment of DBAS and

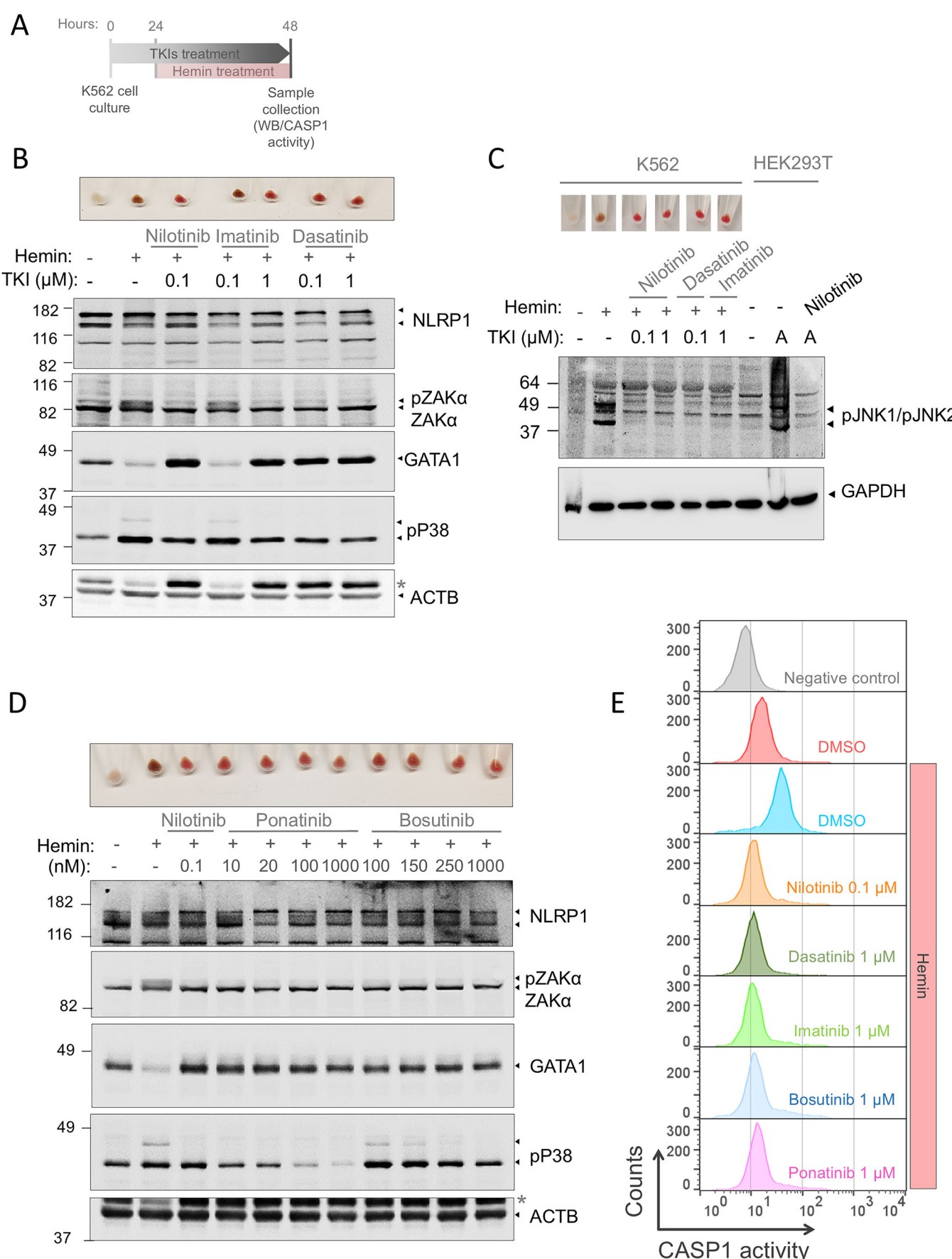

**Figure 5. ZAKα/NLRP1 signaling pathway is activated in K562 cells after erythroid differentiation and its inhibition facilitates terminal erythroid differentiation.**

(A) Chronogram of the experiment. (B–D) K562 cells were pretreated with 0.1 μM nilotinib (B–E), 0.1–1 μM imatinib and 0.1–1 μM dasatinib (B, C), 20–1000 nM ponatinib and 100–1000 bosutinib (D) for 24 h, and then differentiated with 50 μM hemin for another 24 h. Hemoglobin accumulation (A–C) and NLRP1, phosphorylated P38, phosphorylated JNK1/JNK2, GATA1, ZAKα and ACTB/GAPDH (A–D) amounts were then evaluated by Western blot. HEK293T cells were treated for 24 h with 1 μM anisomycin (A) in either the presence or absence of 0.1 μM nilotinib as a control of JNK1/JNK2 activation (C). Immunoblots are representative of three independent experiments. For (B) and (D), GATA1 and ACTB were detected on the same membrane after stripping and reprobing, whereas the remaining proteins were analyzed on separate membranes. *, GATA1. (D) CASP1 activity was measured by FLICA-CASP1 then analyzed by flow cytometry. Representative dot plots at different differentiation times are shown. Source data are available online for this figure.

likely other blood diseases characterized by defective erythropoiesis, such as Fanconi anemia, thalassemia, and myelodysplastic syndromes (Cappellini et al, 2023; Da Costa et al, 2020; Dorenkamp et al, 2023; Guerra et al, 2023) as well as myeloid lineage bias of chronic inflammatory diseases (Marzano et al, 2018; Weiss, 2015), cancer (Wu et al, 2014) and aging (Elias et al, 2017). Of note, although chronic myeloid leukemia (CML) patients treated with TKIs often exhibit anemia, this anemia is primarily attributable to the underlying disease rather than a direct effect of the drugs (Liu et al, 2020). Many CML patients present with anemia at diagnosis—approximately 35% in the chronic phase (Liu et al, 2020)—and early myelosuppression, affecting multiple blood lineages, is common following TKI initiation. This transient cytopenia reflects the suppression of the leukemic clone combined with pre-existing inhibition of normal hematopoiesis by malignant cells, with recovery typically observed once remission is achieved (Steegmann et al, 2016). Furthermore, imatinib is approved for gastrointestinal stromal tumors (GIST), a non-hematologic condition, where severe anemia (grade 3 or 4) occurs at very low rates (approximately 5.4% and 0.7%, respectively) and is likely related to bleeding complications rather than a direct drug effect (https://www.ema.europa.eu/en/documents/product-information/glivec-epar-product-information_en.pdf). These observations underscore that TKI-associated hematologic side effects are transient, dose-dependent, and reversible, further reinforcing their potential as therapeutic agents for congenital anemias.

In summary, the identification of the pivotal role of the NLRP1 inflammasome in hematopoiesis and its activation mechanisms has unveiled a promising therapeutic target for hematopoietic disorders, including congenital anemias and lineage bias disorders. Notably, the substantial efficacy of nilotinib, dasatinib and imatinib in enhancing erythroid differentiation in HSPCs from DBAS patients, as well as in the RPS19-edited human model of DBAS, highlights these drugs as promising emergent treatments for this rare disease. Interestingly, we also observed that low-dose nilotinib increased CFU-GM colonies, consistent with our previous findings in healthy donor cells (Rodriguez-Ruiz et al, 2023). Although the underlying mechanism remains unclear, this effect may reflect off-target activity of nilotinib on other tyrosine kinases, potentially influencing myeloid progenitor expansion independently of the GATA1-mediated lineage switch. This observation underscores the need for further studies to fully elucidate the spectrum of hematopoietic effects of TKIs. Currently, available therapies for DBAS are limited to glucocorticoids, blood transfusions, hematopoietic stem cell transplantation, and gene therapy (Liu and Karlsson, 2024), underscoring the potential impact of TKIs as a novel therapeutic option.

# Methods

## Reagents and tools table

| Reagent/Resource | Reference or Source | Identifier or Catalog Number |
|---|---|---|
| **Experimental models** | | |
| *Zebrafish (Danio rerio), wild type* | Zebrafish International Resource Center | |
| *Tg(lyz:DsRED2)$^{nz50}$* | Hall et al, 2007 | |
| *Tg(gata1a:DsRed)$^{sd2}$* | Traver et al, 2003 | |
| *casper (mitfa$^{w2/w2}$; mpv17$^{a9/a9}$)* | White et al, 2008 | |
| *spint1a$^{hi2217Tg/hi2217Tg}$* | Amsterdam et al, 1999 | |
| **Antibodies** | | |
| CD235A | | 349112, BioLegend, |
| CD71 | | 334108, BioLegend |
| GATA1 | | 13353, Cell Signaling Technology (FC) |
| GATA1 | | 3535, Cell Signaling Technology (WB) |
| NLRP1 | | AF6788, R&D Systems |
| ZAKα | | A301-993A, Bethyl Laboratories |
| pP38 | | MA5-15177, ThermoFisher Scientific |
| pJNK1/pJNK2 | | MA5-15228, ThermoFisher Scientific |
| GAPDH | | ab181602, Abcam |
| ACTB-HRP | | sc-47778, Santa Cruz Biotechnology |
| Sheep IgG | | 31480, Thermofisher Scientific |
| Rabbit IgG | | A6154, Sigma-Aldrich |
| Mouse IgG | | A9044, Sigma-Aldrich |
| **Oligonucleotides and other sequence-based reagents** | | |
| See Tables EV1 and EV3 | | |
| **Chemicals, Enzymes and other reagents** | | |
| Nilotinib | | HY-10159, MedChemExpress |
| Imatinib | | HY-15463, MedChemExpress |
| Dasatinib | | HY-10181, MedChemExpress |
| Bosutinib | | HY-10158, MedChemExpress |

| Reagent/Resource | Reference or Source | Identifier or Catalog Number |
|---|---|---|
| Ponatinib | | HY-12047, MedChemExpress |
| Erythropoietin | | 100-64, Peprotech |
| SCF | | 300-07, PeproTech |
| IL-3 | | 200-03, PeproTech |
| Hemin | | 16009-13-5, Sigma-Aldrich |
| Anysomicin | | HY-18982, MedChemExpress |
| MethoCult™ | | H4431, StemCell Technologies |
| Human AB plasma | | 501973 (SeraCare) |
| Human AB serum | | S40110 (Atlanta Biologicals) |
| Heparin | | H3149, Sigma-Aldrich |
| Insulin | | 9278, Sigma-Aldrich |
| Holo-transferrin | | T0665, Sigma-Aldrich |
| Z-YVAD-AFC | | 688225, Sigma-Aldrich |
| FAM FLICA Caspase-1 kit | | ICT097, Biorad |
| Other | | |
| Human primary CD34+ cells | | SER-CD34-F (ZenBio or 70008 (StemCell Technologies) |

## Zebrafish

Zebrafish (*Danio rerio* H.) were obtained from the Zebrafish International Resource Center and mated, staged, raised and processed as described (Westerfield, 2000). The lines *Tg(lyz:DsRED2)^nz50* (Hall et al, 2007), *Tg(gata1a:DsRed)^sd2* (Traver et al, 2003) and casper (*mitfa^w2/w2; mpv17^a9/a9*) (White et al, 2008) were previously described. The *spint1a^hi2217Tg/hi2217Tg* line was isolated from insertional mutagenesis screens (Amsterdam et al, 1999). The experiments performed comply with the Guidelines of the European Union Council (Directive 2010/63/EU) and the Spanish RD 53/2013. The experiments and procedures performed were approved by the Bioethical Committees of the University of Murcia (approval number #669/2020).

## CRISPR-Cas9 injection in zebrafish

Negative control crRNA (catalog no. 1072544, crRNA standard) and crRNA for *rps19*, *nlrp1* and *zaka* (Table EV1), and tracrRNA were resuspended, duplexed and injected as described previously (Rodriguez-Ruiz et al, 2023). The same amounts of gRNA were used in all experimental groups. The efficiency of each crRNA was tested by amplifying the target sequence with a specific pair of primers (Table EV2) and the amplicon was then analyzed by the TIDE webtool (https://tide.nki.nl/). The edition efficiency obtained averaged ~35% for *rps19*, and ~60% for *nlrp1* and *zaka*, reflecting mosaic editing rather than complete knockout in all embryos. The severity of the phenotype of Rps19-deficient larvae was dose-dependent and could be rescued by co-injection of *rps19* mRNA (Rodriguez-Ruiz et al, 2025).

## Chemical treatments of zebrafish larvae

One dpf larvae were manually dechorionated at 24 hpf and treated for 24 h by bath immersion with the TKIs nilotinib (#HY-10159,

1 μM), imatinib (#HY-15463, 1 μM and 10 μM), dasatinib (#HY-10181, 0.1 μM and 1 μM), bosutinib (#HY-10158, 0.1 μM and 1 μM) and ponatinib (#HY-12047, 0.1 μM and 1 μM), all from MedChemExpress, diluted in egg water supplemented with 0.1% DMSO.

## Imaging of zebrafish larvae

Larvae were anaesthetized in embryo medium with 0.16 mg/ml buffered tricaine and whole-body images were taken with a Leica MZ16F fluorescence stereomicroscope. The number of neutrophils (lyz+) and the number of erythrocytes (mean fluorescence intensity in the heart of Gata1 reporter fish) was determined in blind samples (Tyrkalska et al, 2019).

## Hemoglobin staining

Three dpf zebrafish larvae were anesthetized in buffered tricaine and incubated with 0.62 mg/ml o-dianisidine at room temperature for 15–45 min in the dark. The reaction was monitored under a dissection microscope. Once the staining was completed, they were washed 3 times for 5 min each time with water. The larvae were then fixed for 2 h in 4% methanol-free formaldehyde, were rinsed with PBS containing 0.1% Tween 20 (PBST) three times and stored at 4 °C until imaging with a Leica MZ16F fluorescence stereomicroscope.

## Caspase-1 activity assays

Caspase-1 activity was determined in larval/cell extracts with the fluorometric substrate Z-YVAD 7-Amido-4-trifluoromethylcoumarin (Z-YVAD-AFC, caspase-1 substrate VI, Calbiochem), as previously described (Angosto et al, 2012; Lopez-Castejon et al, 2008; Tyrkalska et al, 2016). Caspase-1 activity in live cells was determined using FAM FLICA Caspase-1 kit (#ICT097, Biorad) following the manufacturer's instructions. Briefly, cells were washed twice with PBS, incubated for 45 min at 37 °C with 1X FAM FLICA in PBS protected from light, washed twice with 1X Apoptosis Wash Buffer, and immediately analyzed by flow cytometry.

## CD34+ cells purification from human cord blood and CRISPR/Cas9 gene editing

CD34+ HSPCs were either enriched by immunomagnetic bead selection from donated human cord blood using an AutoMACS instrument (Miltenyi Biotec) in accordance with the manufacturer's instructions or purchased from ZenBio (#SER-CD34-F) or StemCell Technologies (#70008). Isolated cells were culture at 37 °C in 5% $CO_2$ in HSPC expansion media (StemSpan SFEM II) supplemented with StemSpan CD34+ Expansion Supplement (#02691) (both from StemCell Technologies), with the cell concentration being maintained between $0.2 \times 10^6$ and $1.0 \times 10^6$ cells/ml. Human cord blood samples were provided by Centro de Transfusión de la Comunidad de Madrid under Review Board agreement (SCU IIS FJD-CTCM 2022).

The sgRNAs used for editing *RPS19* and the control gene *AAVS1* have been already described (Bhoopalan et al, 2023). RNP complex was prepared by incubating Cas9-3×NLS protein (alt-rtm s.p. hifi cas9 nuclease v3, IDT) and sgRNA (with 2′-O-methyl 3′ phosphorothioate

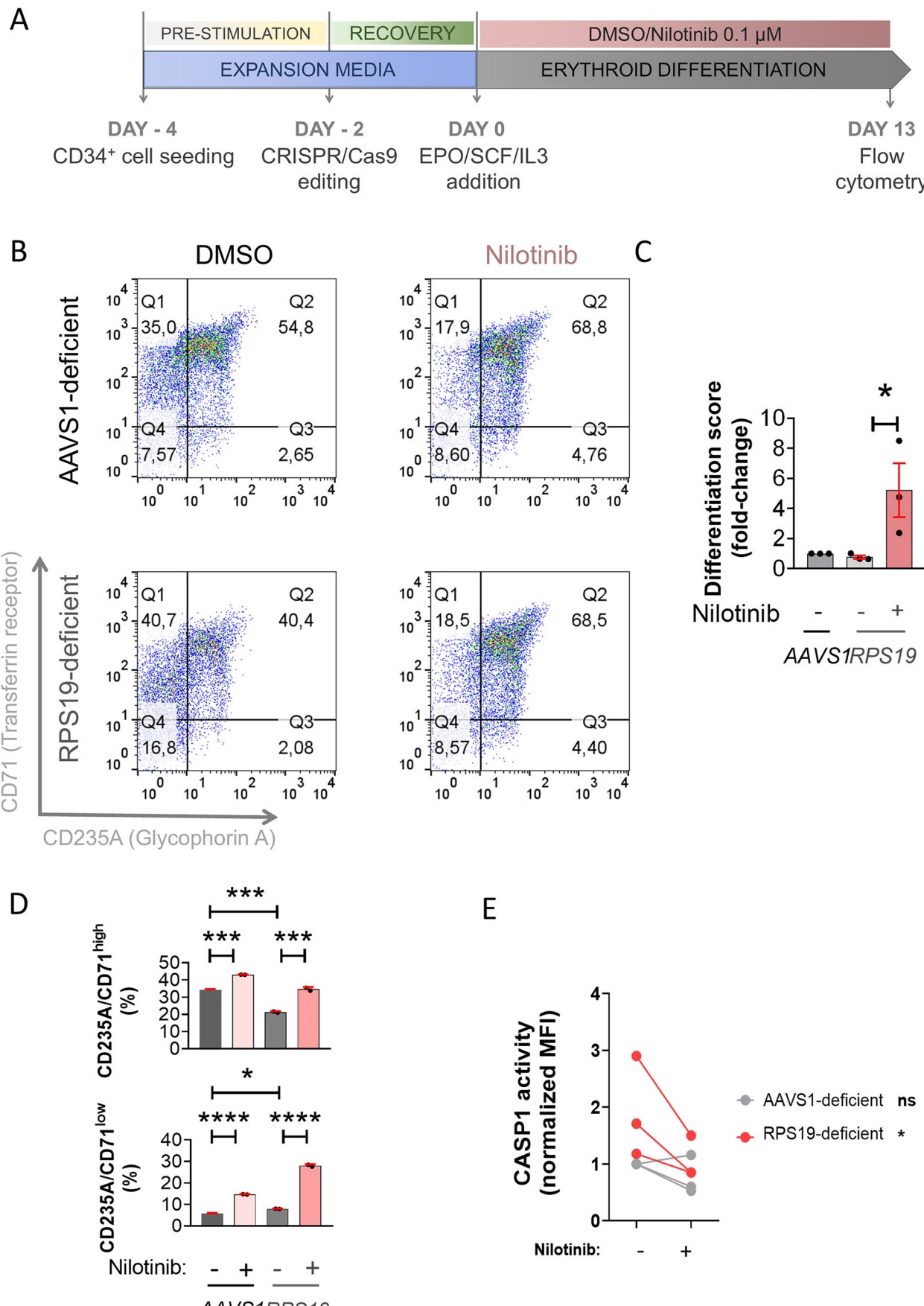

**Figure 6.  Nilotinib alleviates defective erythropoiesis of RPS19-deficient HSPCs.**

(A) Primary human CD34$^+$ cells were purified from human cord blood or purchased from ZenBio or StemCell Technologies, edited with CRISPR/Cas9 and differentiated for 13 days with EPO in the presence of either DMSO or 0.1 µM nilotinib. (B–D) Cells were stained with anti-CD235A-APC (Glycophorin A) and anti-CD71-FITC (Transferrin Receptor), or FAM FLICA, and erythroid differentiation (B–D) and CASP1 activity (E) were then analyzed by flow cytometry. Representative dot plots at different differentiation times are shown in (B). The differentiation score was calculated as the ratio between CD235A$^+$/CD71$^+$ (intermediate erythroid progenitors) and CD235A$^-$/CD71$^+$ (early erythroid progenitors) (C), and the percentage of CD235A$^+$/CD71$^{high}$ (erythroblasts) and CD235A$^+$/CD71$^{low}$ (reticulocytes) (D) and CASP1 activity were determined at 13 days of culture (E). Data are shown as the mean ± SEM (C, E: N = 3, D: N = 2). P values were calculated using one-way ANOVA and Tukey's multiple range test (C, D) or a Student's t-test (E). ns, non-significant; **$p < 0.01$, ***$P < 0.001$ and ****$P < 0.0001$. (C) gRPS19_DMSO wrt gRPS19_NILO $p = 0.0420^*$, (D) Upper graph: gAAVS1_DMSO wrt: gAAVS1_NILO $p = 0.0007^{***}$ or gRPS19_DMSO $p = 0.0002^{***}$, gRPS19_DMSO wrt gRPS19_NILO $p = 0.0001^{***}$, Lower graph: gAAVS1_DMSO wrt: gAAVS1_NILO $p < 0.0001^{****}$ or gRPS19_DMSO $p = 0.0123^*$, gRPS19_DMSO wrt gRPS19_NILO $p < 0.0001^{***}$, (E) DMSO wrt NILO $p = 0.0274^*$. Source data are available online for this figure.

modifications in the first and last 3 nucleotides, Synthego) at a 1:3 molar ratio for 15 min at room temperature. Next, 20,000 HSPCs were washed with PBS and resuspended in P3 buffer (Lonza, catalog V4LP-3002) before RNP was added for a final Cas9 concentration of 0.3 mg/mL and a total volume of 20 µL. Electroporation was performed with a Lonza 4D-nucleofector (#AAF-1003X), using program DZ100, in accordance with the manufacturer's instructions. After RNP treatment, cells were resuspended in HSPC expansion medium and left to recover for 2 days before performing the erythroid differentiation assay (see below).

## Erythroid differentiation assays

K562 cells (CRL-3343; American Type Culture Collection) and HEK293T cells (CRL-11268; American Type Culture Collection) (authenticated by SRS and mycoplasma-free) were maintained and subcultured as described previously (Rodriguez-Ruiz et al, 2023). K562 cells were pre-treated for 24 h with 0.1% DMSO alone or containing the different TKIs and then differentiated for 24 h with 50 µM hemin (#16009-13-5, Sigma-Aldrich) (Smith et al, 2000). In some experiments, HEK293T cells were treated for 24 h with the protein synthesis inhibitor anisomycin (1 µM) (MedChemExpress). Cells were collected at 0 and 24 h post-treatment, washed twice, and stored at −80 °C for further analysis.

Human cord blood CD34$^+$ HSPCs were differentiated in erythroid differentiation medium [IMDM with stabilized glutamine (ThermoFisher Scientific, #12440-061), 2% human AB plasma (SeraCare, #501973), 3% human AB serum (Atlanta Biologicals, #S40110), 1% Pen/Strep (ThermoFisher Scientific), 3 UI/ml heparin (Sigma-Aldrich, #H3149), 10 µg/ml insulin (Sigma-Aldrich, #I9278-5ML), 200 µg/ml holo-transferrin (Sigma-Aldrich, #T0665), 1 IU EPO (Peprotech, # 100-64), 10 ng/ml SCF (PeproTech, #300-07), 1 ng/ml IL-3 (PeproTech, #200-03)] in the presence of DMSO or 0.1 µM nilotinib for 7 days. Human cord blood CD34$^+$ HSPCs at different days of differentiation were stained with antibodies that recognize cell surface markers: CD235A (1:100, #349112, BioLegend) and CD71 (1:100, #334108, BioLegend) to assess erythroid differentiation and GATA1 (1:100 #13353, Cell Signaling Technology), by flow cytometry. All flow cytometry analyses were performed with a FACS Calibur (BD Biosciences) and analyzed with FlowJo software (Tree Star).

## Erythroid/myeloid colony formation assay by mononuclear cells from DBAS patients

Bone marrow aspirates were collected at the Hospital General Universitario José María Morales Meseguer under CEIC approval

number EST: 12/16, while peripheral blood was collected at the Hospital Clínico Universitario Virgen de la Arrixaca under CEIm approval number 2020-6-6-HCUVA. The age and mutations of the patients are summarized in Table EV3. Informed consent was obtained from all subjects or their legal guardians. All procedures involving human participants were conducted in accordance with the ethical standards of the institutional and national research committees, and the principles set out in the WMA Declaration of Helsinki and the Department of Health and Human Services Belmont Report.

BMMCs and PBMCs from DBAS patients were counted and seeded in 6-well plates at 100,000 cells/ml in methylcellulose (MethoCult™ H4431, StemCell Technologies). Cultures were maintained at 37 °C, 5% CO$_2$ and 95% relative humidity. Erythroid (Burst-Forming Unit-Erythroid, BFU-E) and myeloid (Colony-Forming Unit-Granulocytes/Macrophages, CFU-GM) colonies were determined 14 days after cell seeding using morphological criteria.

## Immunoblotting

K562 cells were lysed in 50 mM Tris-HCl (pH 7.5), 150 mM NaCl, 1% (w/v) NP-40 and fresh protease inhibitor (1/20, #P8340, Sigma-Aldrich). Protein quantification was performed with the BCA kit using BSA as standard (Rodriguez-Ruiz et al, 2023). The primary antibodies used were human GATA1 (1:1000, #3535, Cell Signaling), human NLRP1 (1:1000 #AF6788, R&D Systems), human ZAKα (1:1000, #A301-993A, Bethyl Laboratories), human phosphoP38 (1:1000, #MA5-15177, ThermoFisher Scientific), pJNK1/pJNK2 (1:1000 #MA5-15228, Invitrogen), GAPDH (1:2500, #ab181602, Abcam), ACTB-HRP (1:5000, #sc-47778, Santa Cruz Biotechnology). The secondary antibodies used were anti-sheep IgG (1:2500, #31480, Thermofisher), anti-rabbit IgG (1:2500, #A6154, Sigma-Aldrich) and anti-mouse IgG (1:2500, #A9044, Sigma-Aldrich).

## RNA sequencing, bioinformatic analysis and validation by RT–qPCR

Three biological replicates (15 µl at 200 ng/µl) were sent to Novogene for RNA sequencing. Sequencing was performed on an Illumina NovaSeq platform, generating 150 bp paired-end reads. Quality control and differential expression analysis were conducted by the bioinformatics service of the Novogene. Using the data provided, various representations of the differential expression analysis were generated using GraphPad. RT–qPCR was performed as described previously (Rodriguez-Ruiz et al, 2023) and normalized to the ACTB content in each sample using the Pfaffl method (Pfaffl, 2001). The primers used are shown in Table EV2.

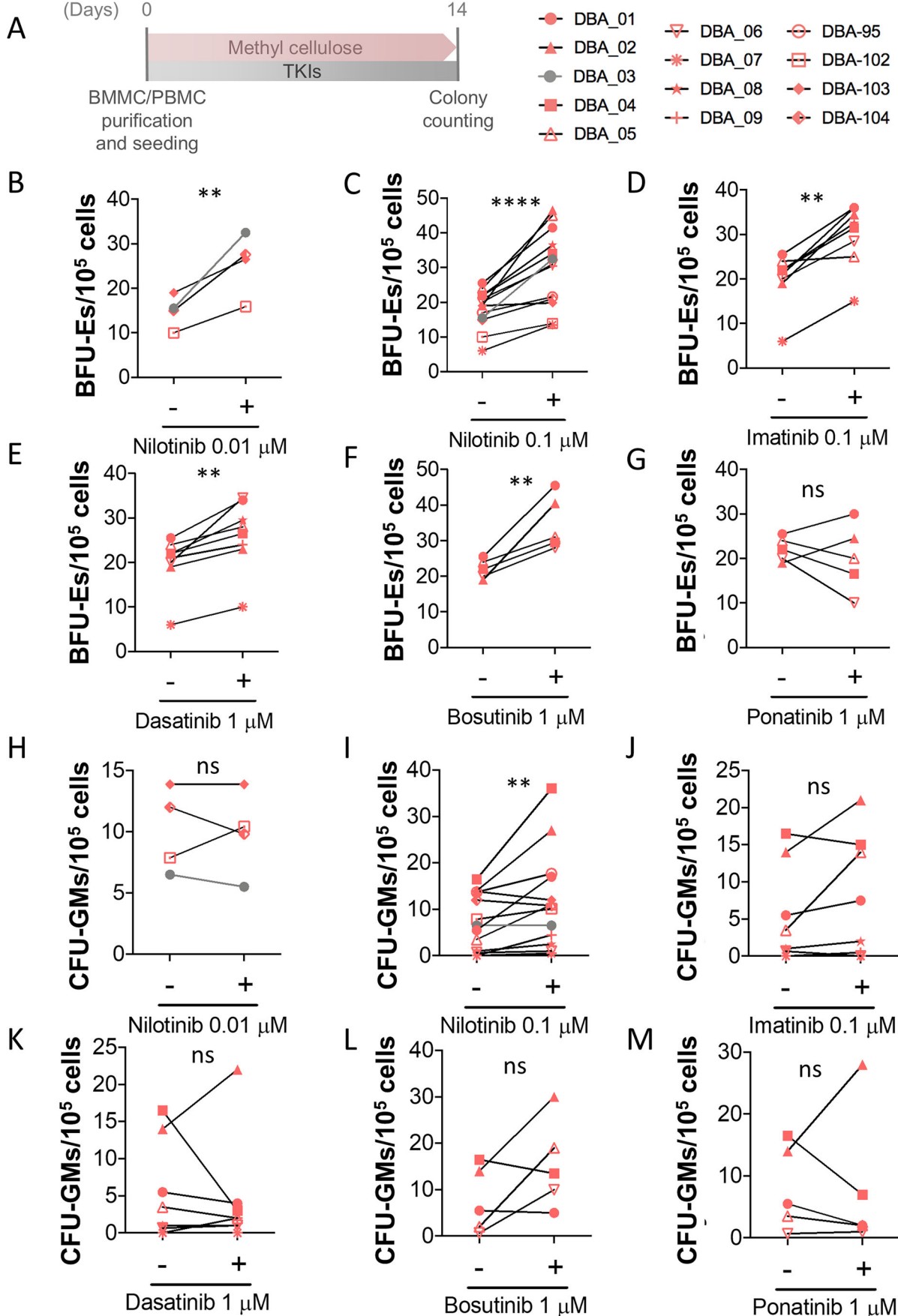

◀ **Figure 7. TKIs alleviate erythropoiesis defects of HSPCs from patients with DBAS.**

(A) Peripheral blood mononuclear cells (PBMCs, red) and bone marrow mononuclear cells (BMMCs, gray) were isolated from 13 patients with DBAS. Cells were cultured for 2 weeks in human methylcellulose complete medium at 37 °C, with or without the indicated TKIs. Burst-forming unit-erythroid (BFU-E) and colony-forming unit-granulocyte macrophage (CFU-GM) colonies were counted based on standard morphological criteria. (B–M) Each dot represents data from an individual patient. The increase in erythroid colonies (BFU-E; B–G) and myeloid colonies (CFU-GM; H, M) upon treatment with TKIs is depicted. Data are shown as the mean ± SEM. ns, non-significant; **$p < 0.01$, ***$P < 0.001$ and ****$P < 0.0001$ according to a Student's $t$-test. (B, C): $p < 0.0001$, (A): $p = 0.009$**, (D): $p = 0.0016$**, (E): $p = 0.0014$**, (F): $p = 0.010$**, (I): $p = 0.0023$**. Source data are available online for this figure.

## The paper explained

### Problem

Diamond-Blackfan anemia syndrome (DBAS) is a rare ribosomopathy characterized by defective red blood cell production. Current treatments rely mainly on corticosteroids or stem cell transplantation, which are limited by toxicity and availability. There is a critical need for safer therapeutic strategies that restore erythropoiesis.

### Results

We demonstrate that selected FDA-approved tyrosine kinase inhibitors (TKIs)—nilotinib, dasatinib, and imatinib—rescue erythropoiesis in several disease-relevant models, including zebrafish, CRISPR-edited human CD34+ hematopoietic progenitors, and cells derived from DBAS patients. Mechanistically, these TKIs inhibit the ZAKα/NLRP1 inflammasome pathway, thereby preventing GATA1 cleavage and restoring erythroid differentiation.

### Impact

Our findings identify a novel therapeutic use for clinically established TKIs in DBAS and related ribosomopathies. By targeting a defined inflammasome-dependent pathway, this drug repurposing approach offers a rapid and realistic route toward clinical translation, with the potential to improve outcomes for patients with congenital anemias.

## Statistical analysis

Embryos were randomly distributed into experimental groups to avoid selection bias and no samples were excluded from the analyses. Data were shown as mean ± SEM. Normality of the data distribution was assessed using the standard tests implemented in GraphPad Prism, selecting the appropriate test according to sample size. Equality of variances was evaluated with the Brown–Forsythe test. Statistical analyses included analysis of variance (ANOVA) followed by Tukey or Bonferroni multiple range tests to determine differences among groups. Differences between two samples were analyzed by Student's $t$-test. At least three independent experiments were performed with zebrafish larvae and biochemical studies. All larvae from the different independent experiments were pooled for plotting and statistical analysis. The total larvae analyzed in each experiment is indicated in all Figures. Three independent caspase-1 activity assays were performed in all experiments using a pool of 30 larvae and one representative experiment is shown with several technical replicates. Colony and erythroid differentiation assays with primary cells were performed with two technical replicates per donor. A log-rank test with the Bonferroni correction for multiple comparisons was used to calculate the statistical differences in the survival of the different experimental groups.

## Data availability

The RNA-seq raw data are available at GEO under accession number GSE300483.

The source data of this paper are collected in the following database record: biostudies:S-SCDT-10_1038-S44321-025-00368-3.

## Peer review information

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

## Acknowledgements

We thank I. Fuentes, P.J. Martínez, M. Ródenas and M.E. Rubio for their excellent technical assistance, and P. Crosier, H. and L.I. Zon for the zebrafish lines. This work was supported by Fundación Séneca, Agencia de Ciencia y Tecnología de la Región de Murcia (research grants 21887/PI/22 and 22242/PDC/23 to VM), MCIN/AEI/10.13039/501100011033 (research grants 2020-113660RB-I00 to VM, and Juan de la Cierva-Incorporación postdoctoral contract to SDT), ISCIII (Miguel Servet CP21/00028 and CP23/00049 to DG-M and SDT, respectively), CIBERER (ER24P7AC7682-ACCI23-12-7682 to AM-L), Consejería de Salud - CARM (ZEBER postdoctoral contract to AM-L), and Diamond-Blackfan Anemia Foundation (USA). The funders had no role in the study design, data collection and analysis, decision to publish, or preparation of the manuscript.

## Author contributions

Juan M Lozano-Gil: Conceptualization; Formal analysis; Investigation; Writing—original draft. Lola Rodríguez-Ruiz: Formal analysis; Investigation. Manuel Palacios: Formal analysis; Investigation. Jorge Peral: Formal analysis; Investigation. Susana Navarro: Formal analysis; Supervision. José L Fuster: Resources; Formal analysis. Cristina Beléndez: Resources; Formal analysis. Andrés Jérez: Resources; Formal analysis. Laura Murillo-Sanjuán: Resources; Formal analysis. Cristina Díaz-de-Heredia: Resources; Formal analysis. Guzmán López-de-Hontanar: Resources; Formal analysis. Josune Zubicaray: Resources; Formal analysis. Julián Sevilla: Resources; Formal analysis. Francisca Ferrer-Marín: Formal analysis. María P Sepulcre: Formal analysis; Supervision. María L Cayuela: Formal analysis; Supervision; Funding acquisition. Diana García-Moreno: Formal analysis; Supervision; Funding acquisition. Alicia Martínez-Lopez: Conceptualization; Formal analysis; Supervision; Funding acquisition. Sylwia D Tyrkalska: Conceptualization; Formal analysis; Supervision; Funding acquisition. Victoriano Mulero: Conceptualization; Formal analysis; Supervision; Funding acquisition; Visualization; Writing—original draft; Project administration; Writing—review and editing.

Source data underlying figure panels in this paper may have individual authorship assigned. Where available, figure panel/source data authorship is listed in the following database record: biostudies:S-SCDT-10_1038-S44321-025-00368-3.

## Disclosure and competing interests statement

A patent for the use of ZAKα inhibitors to treat anemia has been registered by Universidad de Murcia, IMIB Pascual Parrilla, and CIBERER (#P201831288).

# Expanded View Figures

**Figure EV1.** (related to Figs. 2 and 6). **TKIs increased GATA1 protein amount during the erythroid differentiation of primary CD34$^+$ HSPCs.**

(**A**) Primary human CD34$^+$ cells were purified from human cord blood and differentiated for 7 days with EPO in the presence of either DMSO, 100 nM nilotinib, 100 nM imatinib or 1 nM dasatinib. (**B, C**) Cells were stained with anti-GATA1-APC and GATA1$^{high}$ cells analyzed by flow cytometry at 3 and 7 days post-differentiation. Representative dot plots at different differentiation times are also shown. Data are shown as the mean ± SEM. *P* values were calculated using one-way ANOVA and Tukey's multiple range test. All the significant comparisons have a p < 0.0001****. Source data are available online for this figure.

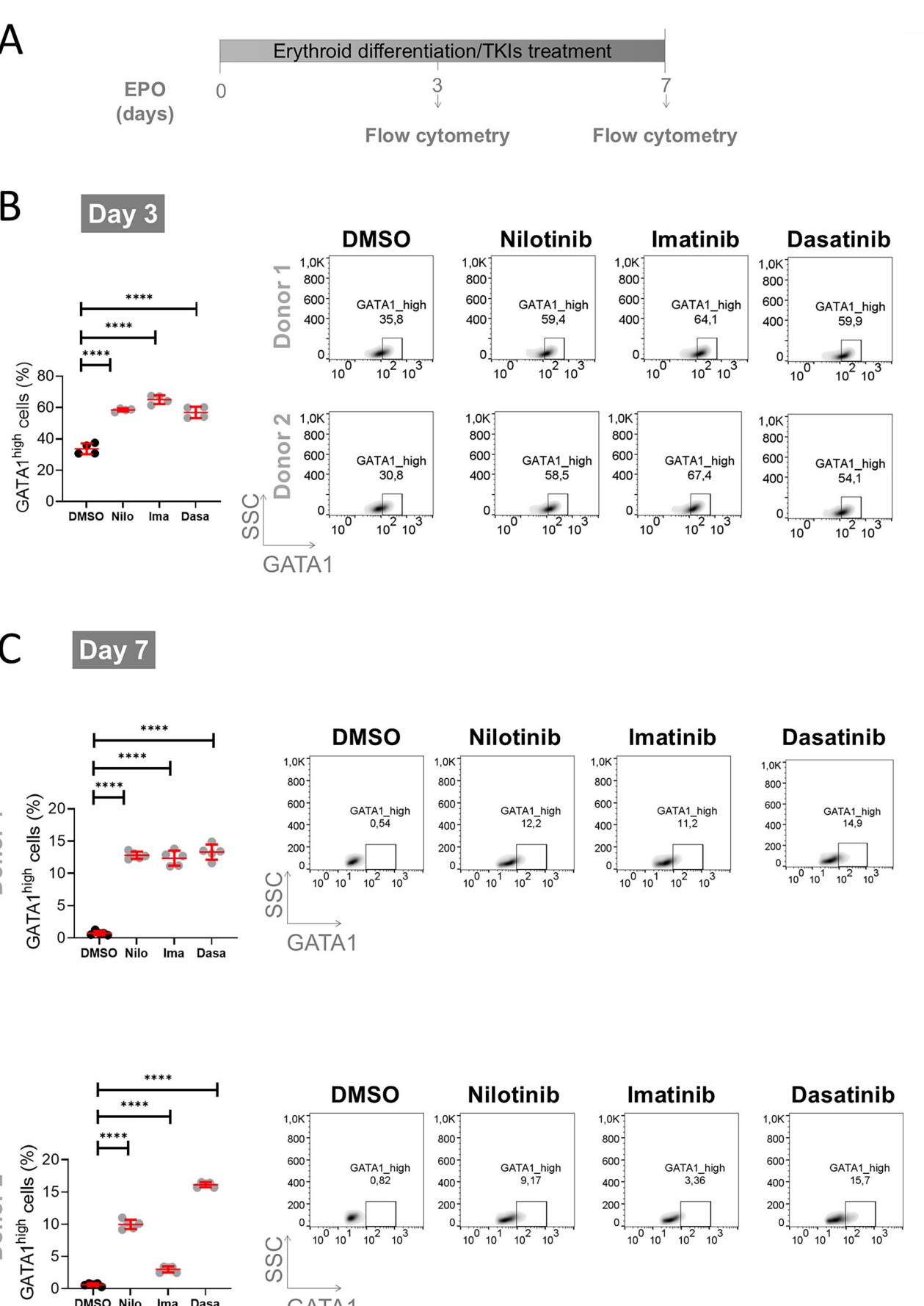

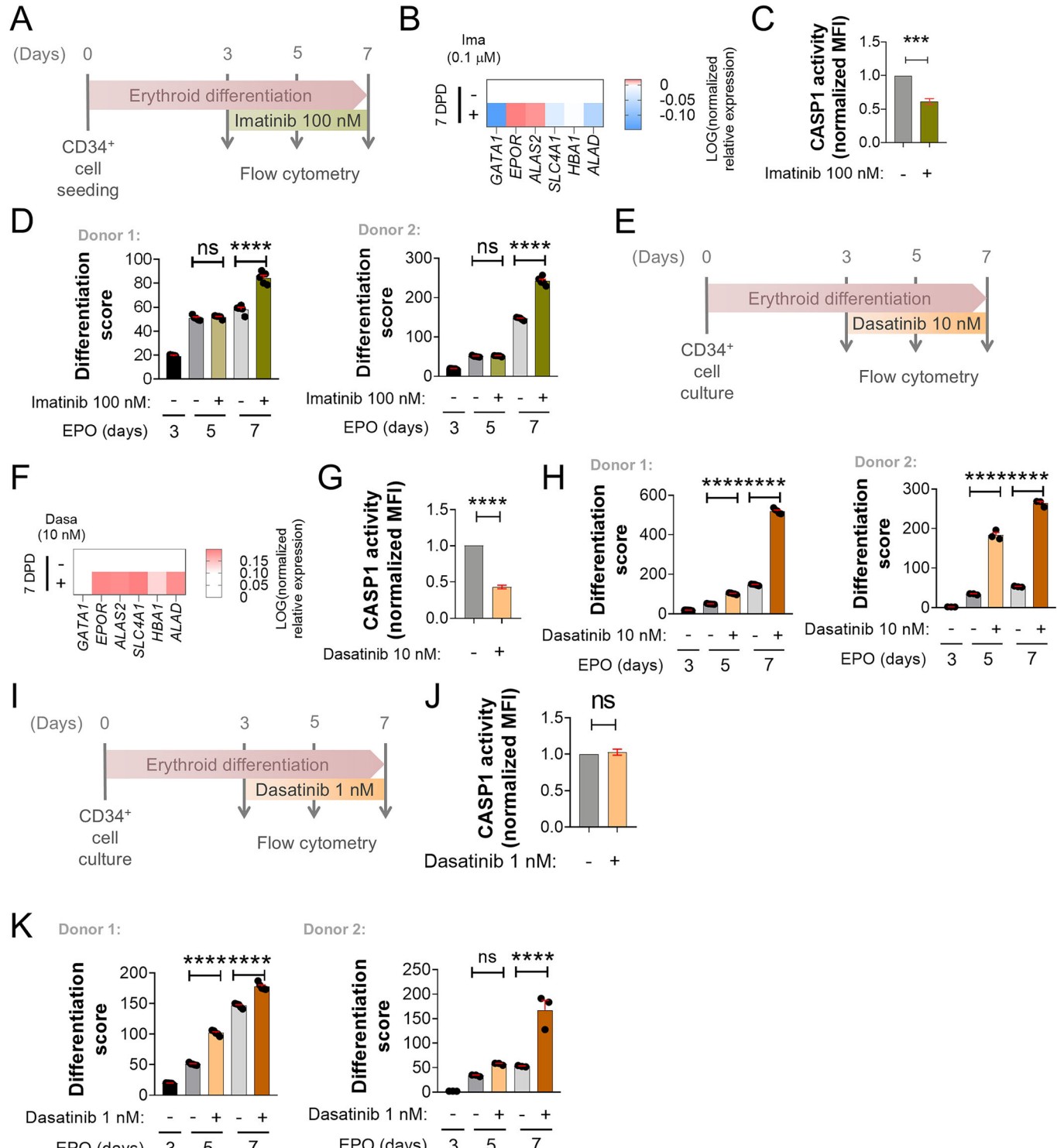

◀ **Figure EV2.** (related to Figs. 2 and 6). **Imatinib and dasatinib replicate the effects of nilotinib on human HSPCs from healthy donors.**

Primary human CD34⁺ HSPCs from healthy donors (ZenBio or StemCell Technologies) were differentiated with EPO in the presence of 0.1 μM imatinib (**A–D**), or 10 (**E–H**) and 1 nM (**I–K**) dasatinib from 3 to 7 days of culture. Cells were stained with anti-CD235A-APC (Glycophorin A) and anti-CD71-FITC (Transferrin Receptor), and erythroid differentiation was then analyzed by flow cytometry. The transcript levels of GATA1-dependent genes (**B**), caspase-1 activity determined with FAM FLICA (**D, G**) and the differentiation score calculated as the ratio between CD235A+/CD71+ (intermediate erythroid progenitors) and CD235A-/CD71+ (early erythroid progenitors) (**D, H, K**) at 7 dpd are shown. Data are shown as the mean ± SEM ($N = 3$). $P$ values were calculated using one-way ANOVA and Tukey's multiple range test (**D, H, K**) or a Student's t-test (**C, G**). ns, non-significant; *$p < 0.05$; **$p < 0.01$; ***$p < 0.01$ and ****$p < 0.0001$. (**C**): $p = 0.0005$***, (**D**): (Donor 1) DMSO_D7 wrt IMA_D7 $p < 0.0001$, (Donor 2) DMSO_D7 wrt IMA_D7 $p < 0.0001$, (**G**): $p < 0.0001$****, (**H**): DMSO_D5 wrt DASA_D5 $p < 0.0001$****, DMSO_D7 wrt DASA_D7 $p < 0.0001$****, (**K**): DMSO_D5 wrt DASA_D5 $p < 0.0001$****, DMSO_D7 wrt DASA_D7 $p < 0.0001$****. Source data are available online for this figure.

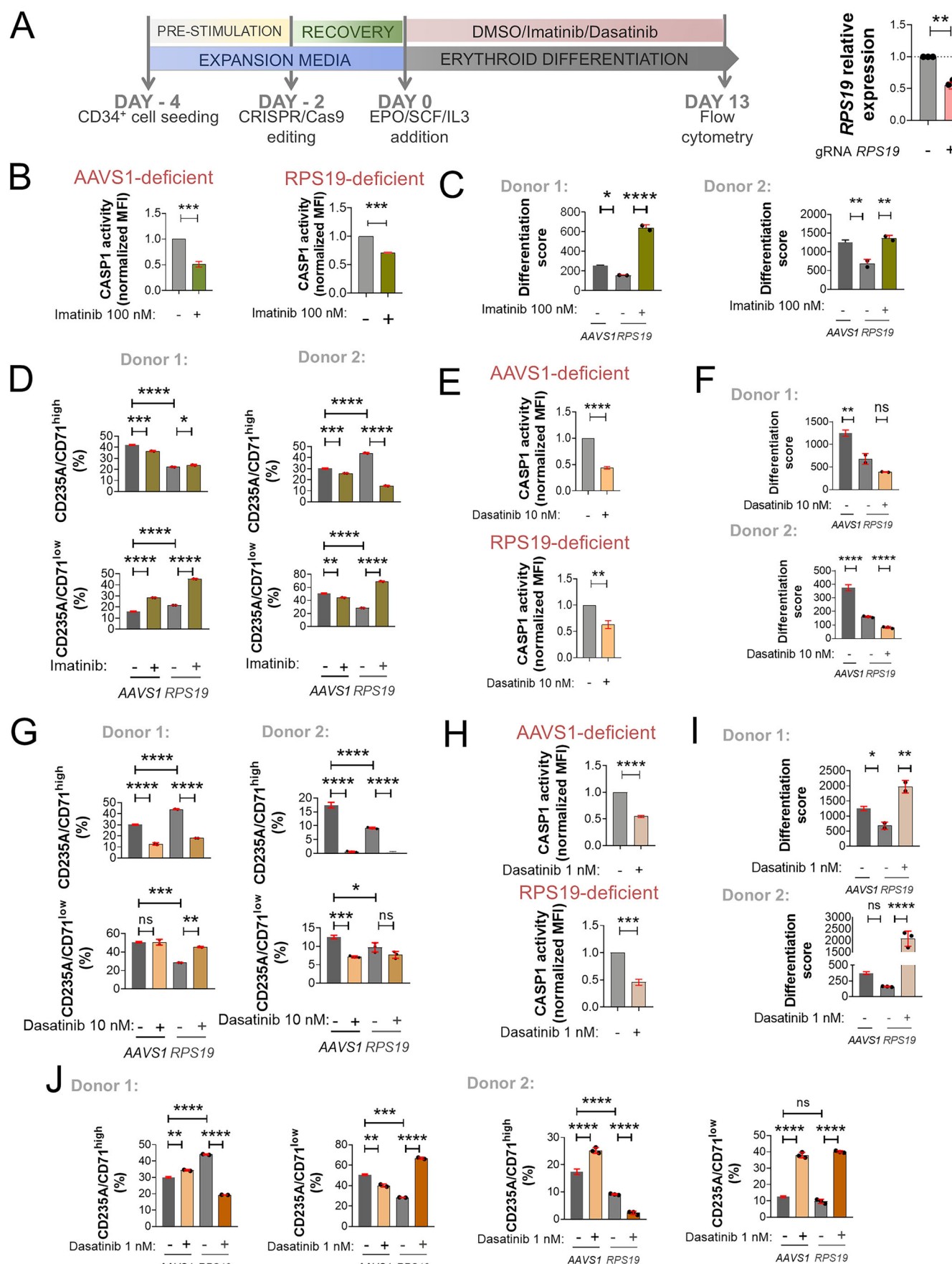

◀  **Figure EV3.  (related to Figs. 2 and 6). Imatinib and dasatinib alleviate defective erythropoiesis of RPS19-deficient HSPCs.**

(A) Primary human CD34+ from healthy donors (ZenBio or StemCell Technologies) were edited with CRISPR/Cas9 and differentiated for 13 days with EPO in the presence of of 0.1 μM imatinib (**A–D**) or 10 nM dasatinib (**E–J**) from 3 to 13 days of culture. Cells were stained with either FAM FLICA or anti-CD235A-APC (Glycophorin A) and anti-CD71-FITC (Transferrin Receptor), and CASP1 activity (**B, H**), erythroid differentiation (**B–D, F, G, I, J**) were then analyzed by flow cytometry. The differentiation score was calculated as the ratio between CD235A + /CD71+ (intermediate erythroid progenitors) and CD235A-/CD71+ (early erythroid progenitors) (**C, F, I**), and the percentage of CD235A + /CD71high (erythroblasts) and CD235A + /CD71low (reticulocytes) (**D, G, J**) at 13 days of culture (**E**). Data are shown as the mean ± SEM ($N = 3$). $P$ values were calculated using one-way ANOVA and Tukey's multiple range test (**C, D, F, G, I, J**) or a Student's $t$-test (**A, B, E, H**). ns, non-significant; *$p < 0.05$; **$p < 0.01$; ***$p < 0.01$ and ****$p < 0.0001$. (**A**): $p = 0.0011$**, (**B**): AAVS1-deficient $p = 0.0009$***, RPS19-deficient $p = 0.0009$***, (**C**): (donor 1) AAVS1_DMSO wrt RPS19_DMSO $p = 0.0271$*, RPS19_DMSO wrt RPS19_IMA $p = 0.0002$***, (donor 2) AAVS1_DMSO wrt RPS19_DMSO $p = 0.0043$**, RPS19_DMSO wrt RPS19_IMA $p = 0.0034$**, (**D**): (Donor 1) (Upper graph) AAVS1_DMSO wrt: AAVS1_IMA $p = 0.0005$*** or RPS19_DMSO $p < 0.0001$****, RPS19_DMSO wrt RPS19_IMA $p = 0.0446$*, (lower graph) all $p < 0.0001$****, (Donor 2) (Upper graph) AAVS1_DMSO wrt: AAVS1_IMA $p = 0.0006$*** or RPS19_DMSO $p < 0.0001$****, RPS19_DMSO wrt RPS19_IMA $p < 0.0001$****, (lower graph) AAVS1_DMSO wrt: AAVS1_IMA $p = 0.0012$** or RPS19_DMSO $p < 0.0001$****, RPS19_DMSO wrt RPS19_IMA $p < 0.0001$****, (**E**): AAVS1-deficient $p < 0.0001$****, RPS19-deficient $p = 0.0076$**, (**F**): (Donor 1) AAVS1_DMSO wrt RPS19_DMSO $p = 0.0012$**, (Donor 2) all significant comparison have $p < 0.0001$****, (**G**): (Donor 1) (upper graph) all significant differences have $p < 0.0001$****, (lower graph) AAVS1_DMSO wrt RPS19_DMSO $p = 0.0006$***. RPS19_DMSO wrt RPS19 $p = 0.0018$**, (Donor 2) (upper graph) all significant differences have $p < 0.0001$****, (lower graph) AAVS1_DMSO wrt: AAVS1_DASA $p = 0.0002$*** or RPS19_DMSO $p = 0.0142$*, (**H**): AAVS1-deficient $p < 0.0001$****, RPS19-deficient $p = 0.0006$***, (**I**): (Donor 1) AAVS1_DMSO wrt RPS19_DMSO $p = 0.0311$*, RPS19_DMSO wrt RPS19_DASA $p = 0.0015$ **, (Donor 2) RPS19_DMSO wrt RPS19_DASA $p < 0.0001$****, (**J**): (Donor 1) (left) AAVS1_DMSO wrt: AAVS1_DASA $p = 0.0012$** or RPS19_DMSO $p < 0.0001$****, RPS19_DMSO wrt RPS19_DASA $p < 0.0001$****, (right) AAVS1_DMSO wrt AAVS1_DASA $p = 0.0015$** or RPS19_DMSO $p = 0.0001$***, RSP19_DMSO wrt RPS19_DASA $p < 0.0001$****, (Donor 2) all significant differences have $p < 0.0001$****. Source data are available online for this figure.

