## [Peer Review File · EMBO Molecular Medicine]

TKI-Mediated Inhibition of NLRP1 Inflammasome Restores Erythropoiesis in DBA Syndrome

Juan Lozano-Gil, Lola Rodriguez-Ruiz, Manuel Palacios, Jorge Peral, Susana Navarro, Jose Fuster, Cristina Belendez, Andres Jerez, Laura Murillo-Sanjuan, Cristina Diaz-de-Heredia, Guzman Lopez-de-Hontanar, Josune Zubicaray, Julian Sevilla, Francisca Ferrer-Marin, María Sepulcre, María L. Cayuela, Diana García-Moreno, Alicia Martinez-Lopez, Sylwia Tyrkalska, and Victoriano Mulero

Corresponding authors: Victoriano Mulero (vmulero@um.es) , Sylwia Tyrkalska (tyrkalska.sylwia@gmail.com), Alicia Martinez-Lopez (alicia.martinez34@um.es)

Review Timeline:

Submission Date:	5th Mar 25
Editorial Decision:	26th Mar 25
Revision Received:	10th Oct 25
Editorial Decision:	11th Nov 25
Revision Received:	20th Nov 25
Editorial Decision:	1st Dec 25
Revision Received:	8th Dec 25
Accepted:	10th Dec 25

Editor: Lise Roth

Transaction Report:

26th Mar 2025

Dear Prof. Mulero,

Thank you for the submission of your manuscript to EMBO Molecular Medicine. We have now received feedback from the three reviewers who agreed to evaluate your manuscript. As you will see from the reports below, the referees acknowledge the interest of the study and are overall supporting publication of your work pending appropriate revisions.

Addressing the reviewers' concerns in full will be necessary for further considering the manuscript in our journal, and acceptance of the manuscript will entail a second round of review. EMBO Molecular Medicine encourages a single round of revision only and therefore, acceptance or rejection of the manuscript will depend on the completeness of your responses included in the next, final version of the manuscript. For this reason, and to save you from any frustrations in the end, I would strongly advise against returning an incomplete revision.

We are expecting your revised manuscript within three to four months, if you anticipate any delay, please contact us.

We require:

4) A .docx formatted letter INCLUDING the reviewers' reports and your detailed point-by-point responses to their comments. As part of the EMBO Press transparent editorial process, the point-by-point response is part of the Review Process File (RPF), which will be published alongside your paper.

5) A complete author checklist, which you can download from our author guidelines (<https://www.embopress.org/page/journal/17574684/authorguide#submissionofrevisions>). Please insert information in the checklist that is also reflected in the manuscript. The completed author checklist will also be part of the RPF.

6) All Materials and Methods need to be described in the main text using our 'Structured Methods' format. According to this format, the Methods section includes a Reagents and Tools Table (listing key reagents, experimental models, software and relevant equipment and including their sources and relevant identifiers) followed by a Methods and Protocols section describing the methods, ideally using a step-by-step protocol format. The aim is to facilitate adoption of the methodologies across labs. Please download and fill our Reagents and Tools Table template (.docx), which you can find in our author guidelines:

<https://www.embopress.org/doi/10.15252/msb.20178071>

7) Please note that all corresponding authors are required to supply an ORCID ID for their name upon submission of a revised manuscript.

8) It is mandatory to include a 'Data Availability' section after the Materials and Methods. Before submitting your revision, primary datasets produced in this study need to be deposited in an appropriate public database, and the accession numbers and database listed under 'Data Availability'. Please remember to provide a reviewer password if the datasets are not yet public (see <https://www.embopress.org/page/journal/17574684/authorguide#dataavailability>).

9) For data quantification: please specify the name of the statistical test used to generate error bars and P values, the number (n) of independent experiments (specify technical or biological replicates) underlying each data point and the test used to calculate p-values in each figure legend. The figure legends should contain a basic description of n, P and the test applied. Graphs must include a description of the bars and the error bars (s.d., s.e.m.). Please provide exact p values.

10) Our journal encourages inclusion of *data citations in the reference list* to directly cite datasets that were re-used and obtained from public databases. Data citations in the article text are distinct from normal bibliographical citations and should directly link to the database records from which the data can be accessed. In the main text, data citations are formatted as follows: "Data ref: Smith et al, 2001" or "Data ref: NCBI Sequence Read Archive PRJNA342805, 2017". In the Reference list, data citations must be labeled with "[DATASET]". A data reference must provide the database name, accession number/identifiers and a resolvable link to the landing page from which the data can be accessed at the end of the reference. Further instructions are available at .

11) We replaced Supplementary Information with Expanded View (EV) Figures and Tables that are collapsible/expandable online. EV Figures should be cited as 'Figure EV1, Figure EV2' etc... in the text and their respective legends should be included in the main text after the legends of regular figures.

12) The paper explained: EMBO Molecular Medicine articles are accompanied by a summary of the articles to emphasize the major findings in the paper and their medical implications for the non-specialist reader. Please provide a draft summary of your article highlighting

13) Author contributions: CRediT has replaced the traditional author contributions section because it offers a systematic machine readable author contributions format that allows for more effective research assessment. Please remove the Authors Contributions from the manuscript and use the free text boxes beneath each contributing author's name in our system to add specific details on the author's contribution. More information is available in our guide to authors.

Please also suggest a visual abstract to illustrate your article as a PNG file 550 px wide x 300-600 px high. A cropped portion of this image will serve as thumbnail for the table of content on our webpage.

16) As part of the EMBO Publications transparent editorial process initiative (see our Editorial at <http://embomolmed.embopress.org/content/2/9/329>), EMBO Molecular Medicine will publish online a Review Process File (RPF) to accompany accepted manuscripts.

In the event of acceptance, this file will be published in conjunction with your paper and will include the anonymous referee reports, your point-by-point response and all pertinent correspondence relating to the manuscript. Let us know whether you agree with the publication of the RPF and as here, if you want to remove or not any figures from it prior to publication. Please note that the Authors checklist will be published at the end of the RPF.

I look forward to receiving your revised manuscript.

Yours sincerely,

Lise Roth

***** Reviewer's comments *****

Referee #1 (Comments on Novelty/Model System for Author):

No concerns here, the authors use a zebrafish model, Crispr'ed human stem cells, and patient-derived cells.

Referee #1 (Remarks for Author):

The study by Gil et al., presents a compelling investigation into the therapeutic potential of FDA approved TKIs for restoring erythropoiesis in DBAS via inhibiting the NLRP1 inflammasome. In a previous EMBO Molecular Medicine paper in 2023, the authors had established a mechanistic link between ribosomal stress-induced activation of the ZAK α /P38/NLRP1/CASP1 axis and defective erythroid differentiation, and showed that the TKI nilotinib promoted erythroid differentiation in K562 cells. As this finding supported the rationale for TKI repurposing in genetic diseases marked by defective erythropoiesis, the authors now expanded their nilotinib findings with additional TKIs, and they employed these TKIs in Diamond-Blackfan Anemia Syndrome (DBAS). By employing a combination of cellular models, zebrafish models, HSPCs, and even DBAS patient-derived cells, the current study highlights the potential of certain TKIs to rescue erythropoiesis in DBAS. This multi-faceted approach not only validates the therapeutic efficacy of TKIs but also offers a promising future strategy for treating DBAS and related ribosomopathies. Although a substantial part of the current manuscript is incremental compared to the authors' previous study (confirming nilotinib findings with other TKIs), the disease-relevant novelty provided by the use of these TKIs in genetic zebrafish and HSPC DBAS models as well as in DBAS patient cells warrants publication of this study, provided addressing the comments below.

Major comments:

1. The data in figure 2 suggest that nilotinib's ability to inhibit caspase-1 and promote erythroid differentiation is more pronounced at day 7 compared to day 10. Could the authors clarify whether this reduced effect is due to a shift in cellular responses as differentiation progresses? As erythroid differentiation involves distinct transcriptional and signaling changes over time, it would be valuable to determine whether specific cellular states are more permissive or resistant to nilotinib's effects. Have the authors investigated whether earlier initiation of treatment (e.g., from day 3) or prolonged exposure (beyond day 10) could sustain its effects more robustly? A time-course experiment measuring caspase-1 activity, differentiation markers, and p38 phosphorylation at additional time points could provide deeper insights into the window of nilotinib's efficacy.
2. Comparing the different TKIs is an important aspect of this study, but the lack of side-by-side dose-response data limits the important ability to compare the relative potency of different inhibitors in order to estimate which may be better for future therapeutic use. Can the authors conduct EC50 measurements for the TKIs to better define their inhibitory potential? Including dose-response data could be performed both in the RPS19-deficient HSPCs and in the Rps19 zebrafish model experiments. This would enhance the translational relevance of the findings, ensuring that the in vivo effects are consistent with cellular models.
3. The partial rescue of DBAS phenotypes in Figure 4G by several TKIs is intriguing but raises questions about the completeness of their therapeutic effects. Is the partial rescue due to limited drug bioavailability, incomplete inhibition of the ZAK α /p38/NLRP1 pathway, or the involvement of compensatory pathways that maintain erythroid defects despite p38 inhibition?

For instance, it would be useful to clarify whether all tested inhibitors exhibit high affinity for ZAK α , as differences in ZAK α binding affinity could account for variability in efficacy. The observation that ponatinib and bosutinib failed to rescue anaemia despite their strong inhibition of the ZAK α /p38 pathway in K562 cells suggests that their effects might be context-dependent. Alternatively, perhaps off-target effects might be contributing to the observed impact of certain TKIs. If available, kinase selectivity data for the inhibitors would help to assess this possibility. Moreover, this information could provide insights into potential threats for unwanted side-effects.

4. While p38 is central to the proposed mechanism, JNK has also been implicated in erythroid differentiation. Do the authors have data on whether JNK activation is altered in this model? Could compensatory activation of JNK in response to p38 inhibition affect differentiation outcomes?

5. The conclusion of Fig 7 that TKIs alleviate defective erythroid differentiation of DBAS patient cells is supported by the observed increase in BFU-E colonies in response to TKIs. However, additional evidence is needed to fully substantiate this claim. Since BFU-E formation reflects an early stage of erythropoiesis, it does not confirm the complete maturation of red blood cells. Further analysis, including flow cytometry for erythroid markers (CD34, CD71, glycophorin A) would provide a more comprehensive evaluation of whether TKIs effectively restore erythropoiesis.

Minor comments:

1. p38 phosphorylation in Figure 5C. The rescue of P-p38 with nilotinib in Figure 5C appears modest. Given that phosphorylation events are often transient, it is possible that an earlier time point would better capture P-p38 dynamics.

2. In Figures 1 and 2 of the manuscript, the authors present compelling and well-supported results demonstrating that nilotinib significantly enhances erythroid differentiation in both K562 cells and human CD34+ HSPCs. However, it would be nice to also include GATA1 protein level data.

3. The sentence on lines 274-276 'In our previous study...' needs a reference.

4. In figures 2D and 6D, it would help the reader when the axes of the plots would be aligned with the CD235/CD71 nomenclature used in the legends.

5. For the readers not familiar with hematopoiesis it might help to mention when starting with the results on line 239 that K562 cells are human bone marrow lymphoblasts.

6. Typo's: several times IL-1B, line 356 NACHT domain, line 74 on the one hand.

Referee #2 (Comments on Novelty/Model System for Author):

Authors use human cells in vitro, and a zebrafish model in vivo, which seems appropriate.

Referee #2 (Remarks for Author):

Lozano Gil and colleagues test TKI inhibitors in a variety of models centred around DBA. In general, they conclude:

- TKIs inhibit caspase-1 activity
- TKIs lead to higher erythropoiesis
- TKIs inhibit ZAK α and thus NLRP1 activation

Connecting the dots here to come to an overall mechanism is logical, however more data and controls would show it more clearly, and discussing/excluding alternative explanations is warranted.

Major points

For the K562 cell model, it would be reasonable to perform additional genetic investigations either by deleting NLRP1, which should enhance differentiation and the TKIs should not have further effect, or overexpressing NLRP1 (with activating mutation?), and confirming that the TKI now has no effect to enhance differentiation.

Quantification of caspase-1 activity using fluorometric substrates is problematic as they can also be activated non-specifically by other caspases. Additional confirmation of Caspase-1 processing by western blot would be ideal.

There is some level of assumption that NLRP1, and not another NLR, is responsible for the activity of the TKIs. Inhibitors of

some NLRs such as MCC950 could be used as negative controls. In addition, where it is not possible to genetically manipulate NLRP1, it may be possible to examine non-denaturing gels blotted for NLRP1, to see if it is oligomerised, and if the TKIs prevent this.

Minor points

The introduction and discussion focus too much on NLRP3 at the expense of relevant NLRP1 literature.

Referee #3 (Comments on Novelty/Model System for Author):

There are some minor questions about the zebrafish experiments and methodology detailed in the response to the authors. The novelty is moderately high because the authors' prior work also assesses models of DBA, ZAKalpha inhibition by Nilotinib and the improvement of differentiation with Nilotinib.

Referee #3 (Remarks for Author):

The authors present data that Nilotinib and other TKIs reduce CASP1 cleavage, and enhance erythroid differentiation in zebrafish models of DBAS and primary patient cells. The data are provocative and compelling - some additional specific questions below. In addition, although somewhat addressed in the discussion, polycythaemia is not a feature of treatment with TKIs in patients and neutropaenia is also uncommon so the mechanisms do not appear to affect steady state haematopoiesis and should be discussed in the final discussion.

Figure 1 - the authors show that nilotinib treatment convincingly leads to erythroid differentiation in K562 cells and upregulates erythroid gene expression. In prior work they have shown that ZAKalpha and p38 are reduced in association with nilotinib treatment and imply this is the mechanism acting here (would be helpful to more clearly specify this in the text).

Figure 2

The addition of nilotinib leads to differentiation and cleavage of CASP1 in primary cord blood CD34 cells. Please specify what question the late addition of nilotinib is attempting to answer vs adding at the beginning. In the text they refer to this process leading to an increase in transcripts of GATA1 but I don't see this data?

Figure 3

It appears that all TKIs tested can inhibit cleavage of CASP1. The initial data on nilotinib indicated this was mediated through ZAKalpha. What is the effect of the other TKIs on ZAKalpha?

3D,E The erythroid counts appear to be on the yolk extension itself? These cells are static not circulating as the other erythroid cells within the vasculature and several authors have suggested may be macrophages with erythroid cells engulfed or macrophage in origin. For clarity in this regard, cells within the vasculature, or the CHT are more appropriate regions to assess erythroid numbers or by flow.

4F Images in F (and G) with Rps19 crRNA have abnormal embryo morphology which improves with the nlrp1 or zaka crRNA. Was this the case? Further, the level of KO is reported in the methods as 35% - does this mean 35% of embryos have any knockdown or all embryos have 35% loss? The images look like more than 50% KO (hets are viable). It's hard to understand either way why the range of Hb content isn't broader in the embryos with Rps19 crRNA, some discussion on this would be helpful.

7 - The trend in patient cells appears to be towards and increase in CFU-GM which is in contrast to the expected findings from zebrafish and the hypothesis that the mechanism is via GATA1 mediated myeloid-erythroid lineage switch. Can the authors speculate on the reason for this.

Referee #1 (Comments on Novelty/Model System for Author):

No concerns here, the authors use a zebrafish model, Crispr'ed human stem cells, and patient-derived cells.

Referee #1 (Remarks for Author):

The study by Gil et al., presents a compelling investigation into the therapeutic potential of FDA approved TKIs for restoring erythropoiesis in DBAS via inhibiting the NLRP1 inflammasome. In a previous EMBO Molecular Medicine paper in 2023, the authors had established a mechanistic link between ribosomal stress-induced activation of the ZAK α /P38/NLRP1/CASP1 axis and defective erythroid differentiation, and showed that the TKI nilotinib promoted erythroid differentiation in K562 cells. As this finding supported the rationale for TKI repurposing in genetic diseases marked by defective erythropoiesis, the authors now expanded their nilotinib findings with additional TKIs, and they employed these TKIs in Diamond-Blackfan Anemia Syndrome (DBAS). By employing a combination of cellular models, zebrafish models, HSPCs, and even DBAS patient-derived cells, the current study highlights the potential of certain TKIs to rescue erythropoiesis in DBAS. This multi-faceted approach not only validates the therapeutic efficacy of TKIs but also offers a promising future strategy for treating DBAS and related ribosomopathies. Although a substantial part of the current manuscript is incremental compared to the authors' previous study (confirming nilotinib findings with other TKIs), the disease-relevant novelty provided by the use of these TKIs in genetic zebrafish and HSPC DBAS models as well as in DBAS patient cells warrants publication of this study, provided addressing the comments below.

We thank the reviewer for the positive and encouraging comments. We appreciate the recognition of the disease-relevant novelty of our study and have addressed all specific points as detailed below.

Major comments:

1. The data in figure 2 suggest that nilotinib's ability to inhibit caspase-1 and promote erythroid differentiation is more pronounced at day 7 compared to day 10. Could the authors clarify whether this reduced effect is due to a shift in cellular responses as differentiation progresses? As erythroid differentiation involves distinct transcriptional and signaling changes over time, it would be valuable to determine whether specific cellular states are more permissive or resistant to nilotinib's effects. Have the authors investigated whether earlier initiation of treatment (e.g., from day 3) or prolonged exposure (beyond day 10) could sustain its effects more robustly? A time-course experiment measuring caspase-1 activity, differentiation markers, and p38 phosphorylation at additional time points could provide deeper insights into the window of nilotinib's efficacy.

We thank the reviewer for this important point. In Figure 2 we specifically show the effect of nilotinib from days 3–7 (panels A–D) and from days 7–10 (panels E–H). In both cases, nilotinib promotes erythroid differentiation and inhibits CASP1 activity, supporting that the inflammasome is active during both early and late stages of differentiation, consistent with our previous Immunity 2019 study. Moreover, we extended these analyses up to 13

days of differentiation in both control and RPS19-deficient CD34⁺ cells and observed consistent results with nilotinib, dasatinib, and imatinib (Fig. 6, Fig. EV2, and Fig. EV3). Regarding time-course analyses: (i) we observed CASP1 inhibition by nilotinib in early (Fig. 2B) and late (Fig. 2F) differentiation, as well as normalization in RPS19-deficient CD34⁺ cells (Fig. 6E) and with dasatinib (Fig. EV2G) and imatinib (Fig. EV3B, EV3E, EV3H); (ii) we analyzed P38 phosphorylation throughout erythroid differentiation, but the results were inconclusive due to the coexistence of two cell populations with high and low pP38 levels. This heterogeneity likely reflects dynamic regulation previously reported in other studies of erythroid differentiation (PMIDs 32429593 & 14694199), and its detailed characterization falls beyond the scope of the present work; and (iii) flow cytometry analysis of GATA1 protein levels showed a significant increase at both day 3 and day 7 with nilotinib, imatinib, and dasatinib, confirming that TKI activity spans different stages of erythroid differentiation through GATA1 stabilization. These novel results on GATA1 protein levels have been included in Fig. EV1 of the revised version.

2. Comparing the different TKIs is an important aspect of this study, but the lack of side-by-side dose-response data limits the important ability to compare the relative potency of different inhibitors in order to estimate which may be better for future therapeutic use. Can the authors conduct EC50 measurements for the TKIs to better define their inhibitory potential? Including dose-response data could be performed both in the RPS19-deficient HSPCs and in the Rps19 zebrafish model experiments. This would enhance the translational relevance of the findings, ensuring that the in vivo effects are consistent with cellular models.

We agree with the reviewer that comparing the potency of different TKIs is an important aspect. The optimal dose of nilotinib was already established in our previous EMBO Molecular Medicine 2023 study and supported by other publications. In the present work, we tested multiple doses of TKIs in zebrafish (Figs. 3C–3F, 4C–4F), K562 cells (Figs. 5B–5D), CD34⁺ cells (Figs. EV1–EV3), and DBAS patient-derived cells (Fig. 7). Importantly, clinical dosing regimens for these TKIs are already well defined for both children and adults, and hematologists will rely on these established safety and exposure data to guide dose selection in future trials. Indeed, in collaboration with Zentiva we are preparing a clinical study with dasatinib, starting with low-dose regimens as recommended by hematologists. Specifically, the Am J Hematol 2022 study (PMID: 36054032) demonstrated that 50 mg/day dasatinib is at least as effective as 100 mg/day while showing a markedly better safety profile (pleural effusion of any grade: 5% vs. 21%). For this reason, we believe that a multiple-dose design is not the most appropriate strategy, and prioritizing patient safety with well-established clinical doses is the best approach. Therefore, we consider that additional in vitro EC50 or multiple-dose analyses in our models would not provide information directly relevant for clinical dosing decisions.

3. The partial rescue of DBAS phenotypes in Figure 4G by several TKIs is intriguing but raises questions about the completeness of their therapeutic effects. Is the partial rescue due to limited drug bioavailability, incomplete inhibition of the ZAKα/p38/NLRP1 pathway, or the involvement of compensatory pathways that maintain erythroid defects despite p38 inhibition? For instance, it would be useful to clarify whether all tested inhibitors exhibit high affinity for ZAKα, as differences in ZAKα binding affinity could account for variability in efficacy. The observation that ponatinib and bosutinib failed to rescue anaemia despite their strong inhibition of the ZAKα/p38 pathway in K562 cells suggests that their effects

might be context-dependent. Alternatively, perhaps off-target effects might be contributing to the observed impact of certain TKIs. If available, kinase selectivity data for the inhibitors would help to assess this possibility. Moreover, this information could provide insights into potential threats for unwanted side-effects.

We thank the reviewer for this insightful comment. The full rescue of anemia observed with genetic inhibition of Nlrp1 or ZAK α (Fig. 4G) suggests that the partial effect of TKIs may be related to the timing of drug treatment (initiated at 24 hpf) and/or to their broader kinase selectivity profiles. Reported data concern the binding affinity of TKIs for ZAK α (nilotinib K_d = 11 nM, dasatinib K_d = 45 nM, imatinib K_d = 2.6 μ M, bosutinib K_d = 80 nM) rather than direct inhibition (<https://www.guidetopharmacology.org/GRAC/ObjectScreenDisplayForward?objectId=2289&familyId=524&screenId=2>), and these affinities do not correlate with the rescue of erythropoiesis observed in vivo. This is consistent with the fact that clinical efficacy does not necessarily correlate with in vitro potency: for example, although dasatinib is ~325-fold more potent than imatinib against BCR::ABL in preclinical assays, the DASISION trial demonstrated that 100 mg/day dasatinib and 400 mg/day imatinib achieve comparable long-term efficacy in CP-CML patients, with dasatinib inducing a faster and deeper molecular response (PMID: 40761712). Thus, potency data alone cannot guide therapeutic dosing decisions. In our study nilotinib, imatinib, and dasatinib consistently rescued DBAS phenotypes across our models, whereas ponatinib and bosutinib did not. Thus, potency data alone cannot guide therapeutic dosing decisions. In our study, nilotinib, imatinib, and dasatinib consistently rescued DBAS phenotypes across models, whereas ponatinib and bosutinib did not, supporting their consideration for repositioning in DBAS. Since dissecting the complex kinase selectivity and pharmacokinetics of each TKI is beyond the scope of the present study, we have focused on demonstrating the robust disease-relevant effects of the most promising candidates.

4. While p38 is central to the proposed mechanism, JNK has also been implicated in erythroid differentiation. Do the authors have data on whether JNK activation is altered in this model? Could compensatory activation of JNK in response to p38 inhibition affect differentiation outcomes?

We thank the reviewer for this insightful question. Since ZAK α is also known to activate JNK, we analyzed JNK1/2 activation during erythroid differentiation of K562 cells. We observed robust phosphorylation of JNK1/2, which was fully blocked by nilotinib, dasatinib, and imatinib (Fig. 5C of the revised version). These results indicate that TKIs prevent both P38 and JNK activation. Importantly, JNK has not been reported to regulate NLRP1 activation downstream of ZAK α (e.g., in keratinocytes exposed to UV light JNK primarily induces apoptosis) (PMIDs: 35857590 & 39591967), and thus no compensatory JNK pathway appears to contribute to erythroid differentiation defects in our model. This has been discussed in the revised version (lines 380-384).

5. The conclusion of Fig 7 that TKIs alleviate defective erythroid differentiation of DBAS patient cells is supported by the observed increase in BFU-E colonies in response to TKIs. However, additional evidence is needed to fully substantiate this claim. Since BFU-E formation reflects an early stage of erythropoiesis, it does not confirm the complete maturation of red blood cells. Further analysis, including flow cytometry for erythroid markers (CD34, CD71, glycophorin A) would provide a more comprehensive evaluation of whether TKIs effectively restore erythropoiesis.

We appreciate this important suggestion. Unfortunately, performing additional flow cytometry analyses on DBAS patient samples is not feasible due to the limited availability of material. However, we note that complete maturation was demonstrated in our RPS19-deficient CD34⁺ model (Figs. 6, EV2 & EV3), which faithfully recapitulates DBAS and supports that TKIs can restore terminal erythroid differentiation.

Minor comments:

1. p38 phosphorylation in Figure 5C. The rescue of P-p38 with nilotinib in Figure 5C appears modest. Given that phosphorylation events are often transient, it is possible that an earlier time point would better capture P-p38 dynamics.

We thank the reviewer for this observation. As previously reported (Rodríguez-Ruiz et al., EMBO Mol Med 2023; Rodríguez-Ruiz et al., HemaSphere 2025), P38 phosphorylation during erythroid differentiation is highly dynamic. In the present study (Fig. 5C), treatment with 0.1 μ M nilotinib clearly inhibited P38 phosphorylation, in agreement with our prior results under ribosomal stress induced by CX-5461. Importantly, the Western blot shows two pP38 bands, with nilotinib fully inhibiting the upper band and partially reducing the lower one; we have now marked both bands in the revised figure to improve clarity.

2. In Figures 1 and 2 of the manuscript, the authors present compelling and well-supported results demonstrating that nilotinib significantly enhances erythroid differentiation in both K562 cells and human CD34⁺ HSPCs. However, it would be nice to also include GATA1 protein level data.

We thank the reviewer for this helpful suggestion. GATA1 protein levels in K562 cells were already shown by Western blot in Fig. 5, where TKIs strongly increased GATA1 expression. In addition, we have now included new flow cytometry data in primary CD34⁺ cells differentiated with EPO, showing a clear increase in GATA1 protein levels with nilotinib, dasatinib, and imatinib at days 3 and 7. These results are presented in Fig. EV1 of the revised manuscript.

3. The sentence on lines 274-276 'In our previous study...' needs a reference.

Fixed.

4. In figures 2D and 6D, it would help the reader when the axes of the plots would be aligned with the CD235/CD71 nomenclature used in the legends.

Fixed.

5. For the readers not familiar with hematopoiesis it might help to mention when starting with the results on line 239 that K562 cells are human bone marrow lymphoblasts.

Fixed.

6. Typo's: several times IL-1B, line 356 NACHT domain, line 74 on the one hand.

Fixed.

Referee #2 (Comments on Novelty/Model System for Author):

Authors use human cells in vitro, and a zebrafish model in vivo, which seems appropriate.

Referee #2 (Remarks for Author):

Lozano Gil and colleagues test TKI inhibitors in a variety of models centred around DBA. In general, they conclude:

- TKIs inhibit caspase-1 activity
- TKIs lead to higher erythropoiesis
- TKIs inhibit ZAKα and thus NLRP1 activation

Connecting the dots here to come to an overall mechanism is logical, however more data and controls would show it more clearly, and discussing/excluding alternative explanations is warranted.

We sincerely thank the reviewer for the thoughtful overall assessment. We have carefully considered all the suggestions and have attempted to address each point raised by incorporating additional experiments, analyses, and clarifications throughout the revised manuscript. We believe these changes have strengthened the mechanistic connection and discussion of alternative explanations, and we are grateful for the reviewer's constructive input.

Major points

For the K562 cell model, it would be reasonable to perform additional genetic investigations either by deleting NLRP1, which should enhance differentiation and the TKIs should not have further effect, or overexpressing NLRP1 (with activating mutation?), and confirming that the TKI now has no effect to enhance differentiation.

We thank the reviewer for this thoughtful suggestion. The genetic approaches proposed would indeed be of interest; however, generating stable NLRP1 mutant or gain-of-function clones and performing the corresponding functional studies would require substantial additional time and go beyond the scope of the present work. Importantly, in our previous studies (Immunity 2019; EMBO Mol Med 2023) we already dissected the mechanism whereby ZAKα activation under ribosomal stress triggers NLRP1 inflammasome assembly, leading to CASP1-mediated cleavage of GATA1. These conclusions were supported by extensive genetic experiments in zebrafish and complementary biochemical and functional assays in K562 cells and CD34⁺ HSPCs.

In the current study, our aim was not to re-establish this mechanism, but to assess the impact of different FDA-approved TKIs on hematopoiesis with a view toward their repositioning for DBA. While we did not perform the exact genetic experiments proposed, we provide additional supportive evidence from two patients carrying potential gain-of-function NLRP1 mutations (p.Trp514Leu and p.Arg1312Gln), diagnosed with severe atopic dermatitis and periodic autoinflammatory syndrome. HSPCs isolated from peripheral blood of these patients displayed a marked defect in erythroid colony formation, which was successfully rescued by nilotinib. These observations further validate the involvement of NLRP1 in hematopoietic regulation and strengthen the translational relevance of our findings.

Figure for reviewers removed

Quantification of caspase-1 activity using fluorometric substrates is problematic as they can also be activated non-specifically by other caspases. Additional confirmation of Caspase-1 processing by western blot would be ideal.

We thank the reviewer for this important observation. Unfortunately, CASP1 protein levels in K562 cells are too low to be reliably detected by Western blot, in contrast to macrophages. This is consistent with the fact that NLRP1 inflammasome activation in HSPCs results in GATA1 cleavage rather than pyroptosis. Nevertheless, we have genetically and biochemically demonstrated the mechanism of NLRP1 activation by ZAK α (via phosphorylation of S107) and the subsequent cleavage of GATA1 at D300 by CASP1 (human) or Caspa (zebrafish) in our previous studies (Immunity 2019; EMBO Mol Med 2023), which strongly supports the specificity of CASP1 activity in this context.

There is some level of assumption that NLRP1, and not another NLR, is responsible for the activity of the TKIs. Inhibitors of some NLRs such as MCC950 could be used as negative controls. In addition, where it is not possible to genetically manipulate NLRP1, it may be possible to examine non-denaturing gels blotted for NLRP1, to see if it is oligomerised, and if the TKIs prevent this.

We appreciate the reviewer's suggestion. Our study builds on detailed gain- and loss-of-function experiments in zebrafish demonstrating that Nlrp1, activated by Zaka under ribosomal stress, is responsible for Gata1 cleavage in HSPCs (Immunity 2019; EMBO Mol Med 2023). These findings were followed by extensive functional and biochemical validation in human models, including K562 cells and RPS19-edited CD34⁺ HSPCs, which confirmed the central role of the ZAK α /NLRP1/CASP1 axis in defective erythropoiesis. The role of NLRP3 was already explored: we tested the selective NLRP3 inhibitor MCC950 in K562 cells and in erythro-myeloid colony assays from healthy donors. In both cases, MCC950 had no effect on erythroid differentiation, in contrast to the caspase-1 inhibitor

VX-765, which significantly increased erythroid colonies. Although the MCC950 results were not included in our EMBO Mol Med 2023 article, they are consistent with our zebrafish data showing that NLRP3 inhibition does not affect hematopoiesis but rather increases neutrophil and macrophage numbers by blocking pyroptosis (bioRxiv 2025.01.22.634381). Taken together, these results strongly support that NLRP1—and not other NLRs—is the relevant inflammasome in this context. Additional MCC950 data are available if the editor considers them necessary.

Minor points

The introduction and discussion focus too much on NLRP3 at the expense of relevant NLRP1 literature.

We thank the reviewer for this comment. We have revised both the Introduction and the Discussion. We agree that it is important to highlight the specific role of NLRP1; however, we also considered it necessary to briefly mention NLRP3 in the opening paragraphs, since until recently it was the only inflammasome reported to regulate hematopoiesis, albeit not HSPC differentiation. After this contextual reference, the text now focuses entirely on NLRP1 and its relevance to our study.

Referee #3 (Comments on Novelty/Model System for Author):

There are some minor questions about the zebrafish experiments and methodology detailed in the response to the authors. The novelty is moderately high because the authors' prior work also assesses models of DBA, ZAKalpha inhibition by Nilotinib and the improvement of differentiation with Nilotinib.

Referee #3 (Remarks for Author):

The authors present data that Nilotinib and other TKIs reduce CASP1 cleavage, and enhance erythroid differentiation in zebrafish models of DBAS and primary patient cells. The data are provocative and compelling - some additional specific questions below. In addition, although somewhat addressed in the discussion, polycythaemia is not a feature of treatment with TKIs in patients and neutropaenia is also uncommon, so the mechanisms do not appear to affect steady state haematopoiesis and should be discussed in the final discussion.

We thank the reviewer for this comment. This issue is already discussed in the manuscript (lines 399–412), where we explain that hematologic side effects observed in TKI-treated patients are largely attributable to the underlying disease (e.g., anemia in CML) or to transient myelosuppression at treatment initiation, rather than to a direct drug effect. Moreover, in non-hematologic settings such as GIST, severe anemia is rare and mainly linked to bleeding complications. Overall, clinical evidence indicates that TKI-associated cytopenias are transient, dose-dependent, and reversible, which supports the potential of these drugs as safe therapeutic candidates for congenital anemias.

1. Figure 1 - the authors show that nilotinib treatment convincingly leads to erythroid differentiation in K562 cells and upregulates erythroid gene expression. In prior work they have shown that ZAKalpha and p38 are reduced in association with nilotinib treatment and

imply this is the mechanism acting here (would be helpful to more clearly specify this in the text)

Thanks for this observation. We have rephrased the text and it now reads: “We first aimed to further investigate the effect of nilotinib, which we previously showed to reduce ZAK α and P38 activation (Rodriguez-Ruiz et al., 2023), on the progression of hemin-induced erythroid differentiation and GATA1 accumulation in K562 cells (Figure 1A)”.

2. Figure 2

The addition of nilotinib leads to differentiation and cleavage of CASP1 in primary cord blood CD34 cells. Please specify what question the late addition of nilotinib is attempting to answer vs adding at the beginning. In the text they refer to this process leading to an increase in transcripts of GATA1 but I don't see this data?

We thank the reviewer for this constructive comment. Our rationale for adding nilotinib both at the onset and at later stages of erythroid differentiation was to determine whether the drug acts primarily on early lineage commitment or can also potentiate erythroid maturation once differentiation is already initiated. This experimental design allows us to distinguish whether nilotinib's effects are limited to early progenitors or extend to cells already engaged in the erythroid program. This has now been clarified in the Results section: “To dissect whether nilotinib acts primarily at the onset of erythroid commitment or can also reinforce differentiation at later stages, we added the drug either at 3-7 days post-differentiation (dpd) or at a later time point (7-10 dpd).”

Regarding GATA1, we agree that in our initial version we did not provide sufficient clarification. In fact, nilotinib does not significantly alter GATA1 mRNA levels, but rather enhances the expression of GATA1-dependent target genes. In response to this and to reviewer 1's related comment, we have now analyzed GATA1 protein expression by flow cytometry, showing that nilotinib, dasatinib, and imatinib robustly increase the percentage of GATA1^{high} cells. These new data are now included in the revised manuscript (Figure EV1).

3. Figure 3

It appears that all TKIs tested can inhibit cleavage of CASP1. The initial data on nilotinib indicated this was mediated through ZAK α . What is the effect of the other TKIs on ZAK α ?

We appreciate the reviewer's insightful question. As suggested, the effect of other TKIs on ZAK α signaling is comparable to that of nilotinib. Specifically, as shown in Figure 5, treatment with dasatinib and imatinib also results in inhibition of P38 and JNK phosphorylation, accumulation of GATA1 protein, reduced CASP1 activity, and promotion of erythroid differentiation in K562 cells. These results indicate that all tested TKIs converge on the same ZAK α /NLRP1 pathway, thereby explaining their common capacity to block CASP1 cleavage.

4. Figure 3D,E The erythroid counts appear to be on the yolk extension itself? These cells are static not circulating as the other erythroid cells within the vasculature and several authors have suggested may be macrophages with erythroid cells engulfed or macrophage in origin. For clarity in this regard cells within the vasculature, or the CHT are more appropriate region to assess erythroid numbers or by flow.

We thank the reviewer for this important observation. To address this concern, we quantified the red fluorescence of erythrocytes not only in the yolk extension but also in the heart, where circulating erythroid cells can be clearly distinguished from static cells. Importantly, this independent analysis yielded comparable results, confirming the increase in erythroid cells upon TKI treatment. These additional data have now been included in the revised version of the figures 3D & 3E.

5. Figure 4F Images in F (and G) with Rps19 crRNA have abnormal embryo morphology which improves with the nlrp1 or zaka crRNA. Was this the case? Further, the level of KO is reported in the methods as 35% - does this mean 35% of embryos have any knockdown or all embryos have 35% loss? The images look like more than 50% KO (hets are viable). Its hard to understand either way why the range of Hb content isn't broader in the embryos with Rps19 crRNA, some discussion on this would be helpful

We thank the reviewer for this excellent observation. Indeed, embryos injected with rps19 gRNA frequently show developmental defects in the head region, a phenotype that mirrors the craniofacial abnormalities described in patients. These defects are associated with increased apoptosis, likely mediated through Jnk activation and independent of Nlrp1. Importantly, they are partially corrected by genetic inhibition of Zaka, suggesting distinct downstream pathways are involved. We are currently pursuing this line of investigation in a separate study.

Regarding the efficiency of CRISPR, we would like to clarify that these are CRISPR F0 crispants rather than stable knockouts. The reported value of ~35% reflects the average editing efficiency across alleles as assessed by sequencing, not that every embryo carries exactly a 35% reduction. Editing outcomes are mosaic and dose-dependent on the gRNA concentration. Importantly, the phenotype can be rescued by co-injection of *rps19* mRNA (please, see Hemasphere 2025), confirming the specificity of the effect. As expected, very high doses of crRNA result in embryonic lethality, consistent with the fact that complete loss of RPS19 is incompatible with survival, whereas patients carry heterozygous mutations. This has been clarified in M&M section and our Hemasphere paper cited.

Finally, we agree with the reviewer that variability in hemoglobin levels among injected embryos might be anticipated. While some degree of variability is observed, the distribution is overall comparable to controls, likely reflecting the mosaic nature of CRISPR editing in crispants rather than uniform knockdown across all cells.

6. Figure 7- The trend in patient cells appear to be towards and increase in CFU-GM which is in contrast to the expected findings from zebrafish and the hypothesis that the mechanisms is via GATA1 mediated myeloid-erythroid lineage switch. Can the authors speculate on the reason for this.

We thank the reviewer for this thoughtful comment. Indeed, we observed that only nilotinib at low doses significantly increased CFU-GM colonies, a finding that is consistent with our previous observations in healthy donor cells (EMBO Mol Med 2023). The underlying reason remains uncertain, but one possibility is that nilotinib may exert additional effects on tyrosine kinases other than ZAK α , thereby modulating myeloid output independently of the GATA1-mediated erythroid–myeloid lineage switch. We have added these sentences to last paragraph of the Discussion: “Interestingly, we also observed that low-dose nilotinib increased CFU-GM colonies, consistent with our previous findings in healthy donor cells (EMBO Mol Med, 2023). Although the underlying mechanism remains

unclear, this effect may reflect off-target activity of nilotinib on other tyrosine kinases, potentially influencing myeloid progenitor expansion independently of the GATA1-mediated lineage switch. This observation underscores the need for further studies to fully elucidate the spectrum of hematopoietic effects of TKIs.”

11th Nov 2025

Dear Prof. Mulero,

Thank you for submitting your revised study. We have now received feedback from the three referees. As you will see below, referees #1 and #2 are satisfied with the revisions, but referee #3 still has concerns about the identification of erythroids vs. macrophages. In a final round of revisions, we would therefore like you to address this remaining point, as well as the following minor editorial issues:

1/ Manuscript text:

- Please accept previous changes and only keep in track changes mode any new modification in the text.
- We can accommodate a maximum of 5 keywords, please adjust accordingly.
- Emails bounced for Manuel Palacios and Jorge Peral (manuel.palacios@ciemat.es and jorge.peral@ciemat.es); please check and correct if needed.
- Methods:
 - o Animals: we note that you referenced previous publications, please make sure you nevertheless sufficient information (origin, housing, CRISPR/Cas9 injection, etc.)
 - o Patients: please include a statement confirming that informed consent was obtained from all subjects and that the experiments conformed to the principles set out in the WMA Declaration of Helsinki and the Department of Health and Human Services Belmont Report.
 - o Antibodies: please provide dilutions/concentrations.
- The Data Availability Section should be moved after the Methods. Provide a URL for the listed dataset, and remove "The rest of datasets generated and/or analyzed during the current study are available within the manuscript."
- Please merge 'Funding' with 'Acknowledgements'. The funding information provided in the manuscript should also be entered in the submission platform.
- Author contributions: CRediT has replaced the traditional author contributions section because it offers a systematic machine readable author contributions format that allows for more effective research assessment. Please remove the Authors Contributions from the manuscript and use the free text boxes beneath each contributing author's name in our system to add specific details on the author's contribution. More information is available in our guide to authors.

2/ Figures:

- The EV figures should be uploaded as individual, high resolution figure files and the legends should be added to the manuscript file (not in the EV figures), under the heading "Expanded View Figure Legends".
- Please make sure all figures/figure panels are referenced in the manuscript text (currently, callouts are missing for Fig. 8 and EV3). There is a callout for Table S1, please correct.
- Fig. 4B/D: please clarify whether the membranes were stripped or include all loading controls.
- Please address the queries from our data editors in the figure legends:
 1. Please note that the legends for figures 2, 4, EV3 is not provided in the sequential manner. This needs to be rectified.
 2. Please note that the legend for figure 2H is missing in the manuscript. This needs to be rectified.
 3. Please define the annotated p values ****/***/**/* as well as provide the exact p-values for the same in the legend of figure EV1 B, C as appropriate.
 4. Please note that the exact p values are not provided in the legends of figures 1C-E; 2B, D, F, H; 3C-E; 4C, D, F, G; 6C-E; 7B, C, D, E, F, I; EV2 C, D, G, H; EV3 A-J.
 5. Please indicate the statistical test used for data analysis in the legends of figures EV1 B, C; EV3 B.
 6. Please note that information related to n is missing in the legends of figures 1C, D, E; 2B, D, F, H; 3C, 6C, D; EV2 C, D, G, H, J, K.

3/ Thank you for providing Source Data, please also complete the Source Data checklist, and provide Source Data for FACS-based figures.

4/ Checklist:

- Ethics/patient photos: please check whether this applies to your study.
- Please check your entries for Data Availability, including the right column.

5/ Please include the Paper Explained in the manuscript text file.

6/ Please suggest a visual abstract to illustrate your article as a PNG file 550 px wide x 300-600 px high. A cropped portion of this image will serve as thumbnail for the table of content on our webpage.

7/ As part of the EMBO Publications transparent editorial process initiative (see our Editorial at <http://embomolmed.embopress.org/content/2/9/329>), EMBO Molecular Medicine will publish online a Review Process File (RPF) to accompany accepted manuscripts.

This file will be published in conjunction with your paper and will include the anonymous referee reports, your point-by-point response and all pertinent correspondence relating to the manuscript. Let us know whether you agree with the publication of the RPF and as here, if you want to remove or not any figures from it prior to publication. Please note that the Authors checklist will be published at the end of the RPF.

I look forward to receiving your revised manuscript.

Yours sincerely,

Lise Roth

***** Reviewer's comments *****

Referee #1 (Comments on Novelty/Model System for Author):

No concerns here, the authors use a zebrafish model, Crispr'ed human stem cells, and patient-derived cells. Novelty a bit incremental as compared to the authors' previous 2019 Immunity and 2023 EmboMolMed papers.

Referee #1 (Remarks for Author):

The authors have adequately answered my questions and I now support publication of this paper in EmboMolMed. However, I would recommend to include the interesting data with the NLRp1 GOF patient cells shown in response to reviewer 2 in the manuscript, as well as the MCC950 controls mentioned in response to reviewer 2.

Referee #2 (Comments on Novelty/Model System for Author):

As previously stated

Referee #2 (Remarks for Author):

My comments were addressed

Referee #3 (Comments on Novelty/Model System for Author):

Overall the data presented is exciting and builds on existing findings published by the group. I am concerned however, that they have not addressed a fundamental question around the model that I brought up in review. I feel this needs to be further addressed.

Referee #3 (Remarks for Author):

The majority on my queries have been answered. However, regarding Figure 3 D and E the authors have rather glossed over the concern that they are counting macrophages and not red cells. I acknowledge the MFI of the heart adds weight the findings that erythroid cells are increased since these are circulating erythroid cells, but the dotted line shown delineating what has been counted on the yolk extension, and the arrows all point at static cells. I am not aware of a paper that uses these cells to define erythroid cells in the Tg(gata1:dsRed) line as these cells are larger than the circulating erythroid cells and static on the yolk or within fat of the yolk. They have features of macrophages. Further, in situ hybridisation of Gata1a, band3, globin genes and other specific erythroid genes do not show any expression in this region. The red cells are circulating or in the CHT and the dotted line excludes the circulating cells in the aorta. I do not think it is legitimate to use this analysis as showing an increase in erythroid cells.

Referee #1 (Comments on Novelty/Model System for Author):

No concerns here, the authors use a zebrafish model, Crispr'ed human stem cells, and patient-derived cells. Novelty a bit incremental as compared to the authors' previous 2019 Immunity and 2023 EmboMolMed papers.

Referee #1 (Remarks for Author):

The authors have adequately answered my questions and I now support publication of this paper in EmboMolMed. However, I would recommend to include the interesting data with the NLRp1 GOF patient cells shown in response to reviewer 2 in the manuscript, as well as the MCC950 controls mentioned in response to reviewer 2.

We thank the reviewer for their positive assessment and support for publication. We fully agree that the data from patients carrying NLRP1 gain-of-function variants are of great interest. However, these results are part of ongoing studies aimed at the detailed molecular characterization of the mutants, and public release of these preliminary data could compromise our future work. For this reason, we prefer not to include them in the present manuscript.

Regarding MCC950, we discussed these control experiments in our previous response, but did not include them because they did not contribute additional mechanistic information. As our study focuses specifically on the ZAK α -NLRP1 axis, and not on NLRP3, the MCC950 data are not directly relevant to the central pathway investigated in this work.

Referee #2 (Comments on Novelty/Model System for Author):

As previously stated

Referee #2 (Remarks for Author):

My comments were addressed

We thank the reviewer for their careful evaluation and appreciate that all comments were considered satisfactorily addressed in the revised version.

Referee #3 (Comments on Novelty/Model System for Author):

Overall the data presented is exciting and builds on existing findings published by the group. I am concerned however, that they have not addressed a fundamental question around the model that I brought up in review. I feel this needs to be further addressed.

Referee #3 (Remarks for Author):

The majority on my queries have been answered. However, regarding Figure 3 D and E the authors have rather glossed over the concern that they are counting macrophages and not red cells. I acknowledge the MFI of the heart adds weight the findings that erythroid cells are increased since these are circulating erythroid cells, but the dotted line shown delineating what has been counted on the yolk extension, and the arrows all point at static cells. I am not aware of a paper that uses these cells to define erythroid cells in the Tg(gata1:dsRed) line as these cells are larger than the

circulating erythroid cells and static on the yolk or within fat of the yolk. They have features of macrophages. Further, in situ hybridisation of Gata1a, band3, globin genes and other specific erythroid genes do not show any expression in this region. The red cells are circulating or in the CHT and the dotted line excludes the circulating cells in the aorta. I do not think it is legitimate to use this analysis as showing an increase in erythroid cells.

We thank the reviewer for raising this important point. In response, we have removed this analysis from the revised manuscript (Figs. 3D & 3E). We now focus exclusively on circulating erythroid cells in the heart, where quantitative fluorescence measurements show a clear increase upon TKI treatment, supporting our conclusion. This revised approach avoids the ambiguity associated with yolk-associated DsRed⁺ cells and aligns with the established and reviewer-noted localization of erythroid cells in zebrafish embryos.

1st Dec 2025

Dear Prof. Mulero,

Thank you for submitting your revised files. There are still a few editorial matters that need to be addressed before I can accept your manuscript:

1/ Methods:

- as per our guidelines for authors (<https://www.embopress.org/page/journal/17574684/authorguide#researcharticleguide>), this section should contain sufficient detail so that all experimental procedures can be repeated by others, in conjunction with cited references. In the Zebrafish and CRISPR-Cas9 sections, some references are not listed or accessible (Westerfield, 2000; Amsterdam, Burgess et al., 1999). Please make sure you provide sufficient details directly in the methods text, and check the cited references.

- Patients: please include the full statement confirming that the experiments conformed to the principles set out in the WMA Declaration of Helsinki and the Department of Health and Human Services Belmont Report.

- Antibodies: please provide dilutions/concentrations.

- Data Availability: please provide a URL for the listed dataset.

2/ Please enter the funding information listed in the Acknowledgements in the submission platform.

3/ Figures legends:

Please address the queries from our data editors in the figure legends: the legends for figures 2, 4, EV3 is not provided in a sequential manner. This needs to be rectified (you may relabel the different panels if that makes it easier).

4/ Thank you for providing the Source Data files and the checklist. In the checklist, please ensure that you have checked all panels for which SD have been provided. Please indicate in the right column the SD category (numerical data, microscopy images, etc). Please check again the SD provided (for instance, SD are provided for Figure 2H, but there is no panel 2H).

Thank you for addressing these remaining issues. I look forward to receiving your revised manuscript at your earliest convenience.

Yours sincerely,

Lise Roth

The authors addressed the remaining editorial issues.

10th Dec 2025

Dear Prof. Mulero,

Thank you for submitting your revised files. I am pleased to inform you that your manuscript is accepted for publication and is now being sent to our publisher to be included in the next available issue of EMBO Molecular Medicine!

You may qualify for financial assistance for your publication charges - either via a Springer Nature fully open access agreement or an EMBO initiative. Check your eligibility: <https://link.springer.com/journal/44321/how-to-publish-with-us>

With kind regards,

Lise

>>> Please note that it is EMBO Molecular Medicine policy for the transcript of the editorial process (containing referee reports and your response letter) to be published as an online supplement to each paper. If you do NOT want this, you will need to inform the Editorial Office via email immediately. More information is available here: <https://link.springer.com/partners/embo-press/editorial-policies#Peer%20review>